# Pivoting Factorization: A Compact Meta Low-Rank Representation of Sparsity for Efficient Inference in Large Language Models

**Jialin Zhao** [1 2]   **Yingtao Zhang** [1 2]   **Carlo Vittorio Cannistraci** [1 2 3]

## Abstract

The rapid growth of Large Language Models has driven demand for effective model compression techniques to reduce memory and computation costs. Low-rank pruning has gained attention for its GPU compatibility across all densities. However, low-rank pruning struggles to match the performance of semi-structured pruning, often doubling perplexity at similar densities. In this paper, we propose Pivoting Factorization (**PIFA**), a novel **lossless** meta low-rank representation that unsupervisedly learns a **compact** form of any low-rank representation, effectively eliminating redundant information. PIFA identifies pivot rows (linearly independent rows) and expresses non-pivot rows as linear combinations, achieving **24.2%** additional memory savings and **24.6%** faster inference over low-rank layers at rank = 50% of dimension. To mitigate the performance degradation caused by low-rank pruning, we introduce a novel, retraining-free reconstruction method that minimizes error accumulation (**M**). **MPIFA**, combining M and PIFA into an end-to-end framework, significantly outperforms existing low-rank pruning methods, and achieves performance comparable to semi-structured pruning, while surpassing it in GPU efficiency and compatibility. Our code is available at https://github.com/biomedical-cybernetics/pivoting-factorization.

## 1. Introduction

The rapid growth of Large Language Models (LLMs) (Radford, 2018; Radford et al., 2019; Mann et al., 2020; Touvron et al., 2023a) has revolutionized natural language processing tasks but has also introduced significant challenges related to memory consumption and computational costs. Deploying these models efficiently, particularly on resource-limited hardware, has driven a surge of interest in model compression techniques (Wan et al., 2023; Zhu et al., 2024). Among these techniques, *semi-structured pruning*, specifically N:M sparsity, has emerged as a promising approach due to its hardware-friendly nature, enabling efficient acceleration on NVIDIA's Ampere GPUs (Mishra et al., 2021; nvi, 2020). However, semi-structured pruning suffers from two major limitations: it is restricted to specific hardware architectures, and it couldn't flexible adjust density.

In contrast, *low-rank pruning* methods, primarily based on Singular Value Decomposition (SVD), preserve the coherence of tensor shapes, making them universally compatible with any GPU architecture at any density. Recent works (Yuan et al., 2023; Wang et al., 2024) have demonstrated the potential of low-rank decomposition in compressing LLMs. However, despite their flexibility, these methods struggle to compete with semi-structured pruning in terms of performance, often resulting in a **2x increase in perplexity (PPL)** at the same densities. This performance gap stems primarily from two challenges: (1) Low-rank pruning introduces **information redundancy** in the decomposed weight matrices, and (2) existing reconstruction methods **accumulate errors** across layers, leading to suboptimal performance.

To address challenge (1), we propose **Pivoting Factorization (PIFA)**, a novel lossless low-rank representation that eliminates redundancy and enhances computational efficiency. PIFA is a meta low-rank representation, because it unsupervisedly learns a compact representation of any other learned low-rank representation. PIFA identifies $r$ linearly independent rows from a singular weight matrix, which we refer to as pivot rows, and represents all other rows as linear combinations of these pivot rows. PIFA achieves significant improvements in both speedup and memory reduction during inference. Specifically, at $r/d = 0.5$, the PIFA layer achieves **24.2%**

[1]Center for Complex Network Intelligence (CCNI), Tsinghua Laboratory of Brain and Intelligence (THBI), Department of Psychological and Cognitive Sciences [2]Department of Computer Science [3]Department of Biomedical Engineering, Tsinghua University, China. Correspondence to: Jialin Zhao <jialin.zhao97@gmail.com>, Carlo Vittorio Cannistraci <kalokagathos.agon@gmail.com>.

*Proceedings of the $42^{nd}$ International Conference on Machine Learning*, Vancouver, Canada. PMLR 267, 2025. Copyright 2025 by the author(s).

Table 1: **Comparison of PIFA with other sparsity.**

Figure 1: **Comparison of parameter ratios**.

| Method | CPU Speedup | GPU Speedup | GPU Mem Reduction | Any Sparsity | GPU Support | Performance |
|---|---|---|---|---|---|---|
| Unstructured Sparsity | ✓ | ✗ | ✗ | ✓ | ✗ | ✓✓✓ |
| Semi-Structured Sparsity | ✓ | ✓ | ✓ | ✗ | Ampere GPU | ✓✓ |
| Structured Sparsity | ✓ | ✓ | ✓ | ✓ | General | ✓ |
| SVD-Based Low-Rank Sparsity | ✓ | ✓ | ✓ | ✓ | General | ✓ |
| PIFA Low-Rank Sparsity | ✓ | ✓ | ✓ | ✓ | General | ✓✓ |

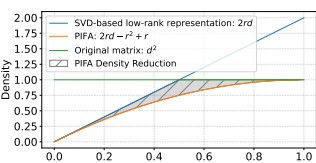

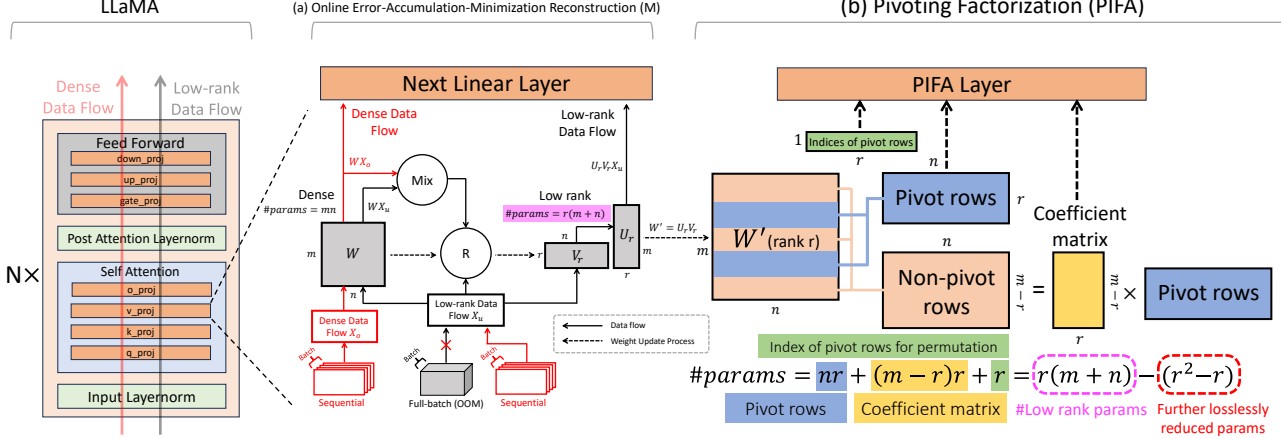

Figure 2: Illustration of the low-rank pruning method **MPIFA** (Algorithm 3), which consists of: **(a) Online Error-Accumulation-Minimization Reconstruction (M).** Block $R$ solves the least-squares optimization problem. The improvements upon SVD-LLM's full-batch reconstruction, highlighted in red, include using the dense data flow to minimize error accumulation, and processing each sample sequentially to avoid GPU memory overflow. **(b) Pivoting Factorization (PIFA).** For **any singular matrix** with rank $r$, Pivoting Factorization further reduces $r^2 - r$ parameters, with no additional loss induced.

additional memory savings and **24.6%** faster inference compared to SVD-based low-rank layers, **without inducing any loss**.

To address challenge (2), we propose an **Online Error-Accumulation-Minimization Reconstruction (M)** algorithm that minimizes error accumulation—a problem pervasive in existing reconstruction methods for both low-rank pruning (Wang et al., 2024) and semi-structured pruning (Frantar & Alistarh, 2023; Li et al., 2024). Existing methods rely on **degraded data flow**, where accumulated errors from previous modules propagate through the reconstruction process, leading to suboptimal performance. Our approach addresses this issue by combining dense data flow and low-rank data flow, as reconstruction targets, effectively mitigating the errors carried forward from earlier layers. Furthermore, the method operates online, processing large numbers of calibration samples sequentially to stay within GPU memory constraints.

Combining M and PIFA, we present **MPIFA**—an end-to-end, retraining-free low-rank pruning framework for LLMs. MPIFA significantly outperforms existing low-rank pruning methods, reducing the perplexity gap by **40%-70%**

on LLaMA2 models (7B, 13B, 70B) and the LLaMA3-8B model [1]. Further experiments show that MPIFA consistently achieves **superior** speedup and memory savings both layer-wise and end-to-end compared to *semi-structured pruning*, while maintaining **comparable** or even better perplexity. Notably, as shown in Table 6, at $d = 32768$, PIFA with 55% density achieves a **2.1×** speedup, whereas various implementations of semi-sparse methods are either slower than the dense linear layer or fail to execute.

The two main contributions of this work can be summarized as follows:

1. We propose **Pivoting Factorization (PIFA)**, a novel **lossless** meta low-rank representation that unsupervisedly learns a **compact** form of any SVD-based low-rank representation, effectively compressing out redundant information.

2. We introduce an **Online Error-Accumulation-Minimization Reconstruction (M)** algorithm that mitigates error accumulation by leveraging multiple data flows for reconstruction.

---

[1] https://www.llama.com/

## 2. Related Work

### 2.1. Connection-wise pruning

**Pruning methods** We define connection-wise pruning as removing certain connections between neurons in the network that are deemed less important. To achieve this, a series of methods have been proposed. Optimal Brain Damage (OBD) (Le Cun et al., 1990) and Optimal Brain Surgeon (OBS) (Hassibi et al., 1993) were proposed to identify the weight saliency by computing the Hessian matrix using calibration data. Recent methods such as SparseGPT (Frantar & Alistarh, 2023), Wanda (Sun et al., 2024), RIA (Zhang et al., 2024), along with other works (Fang et al., 2024; Dong et al., 2024; Das et al., 2024), have advanced these ideas. Wanda prunes weights with the smallest magnitudes multiplied by input activations. Relative Importance and Activations (RIA) jointly considers both the input and output channels of weights along with activation information. Furthermore, OWL (Yin et al.) explores non-uniform sparsity by pruning based on the distribution of outlier activations, while other works (Lu et al., 2024; Mocanu et al., 2018; Ye et al., 2020; Zhuang et al., 2018) investigate alternative criteria for non-uniform sparsity.

**Pruning granularity** (compared in Table 1):

1. **Unstructured pruning** removes individual weights based on specific criteria. Today, unstructured pruning is a critical technique for compressing large language models (LLMs) to balance performance and computational efficiency. However, unstructured pruning can only accelerate computations on CPUs due to its unstructured sparsity pattern.

2. **Semi-structured pruning**, i.e., N:M sparsity, enforces that in every group of $M$ consecutive elements, $N$ must be zeroed out. This constraint is hardware-friendly and enables optimized acceleration on GPUs like NVIDIA's Ampere architecture (Mishra et al., 2021). However, semi-structured pruning is constrained by its sparsity pattern, preventing flexible density adjustments and making it inapplicable for acceleration on general GPUs.

3. **Structured pruning** (Ma et al., 2023; van der Ouderaa et al., 2024; Ashkboos et al., 2024; Lin et al., 2024; Song et al., 2024; Men et al., 2024; Gao et al., 2024) removes entire components of the model, such as neurons, channels, or attention heads, rather than individual weights. This method preserves tensor alignment and coherence, ensuring compatibility with all GPUs and enabling significant acceleration on both CPUs and GPUs. However, in LLMs, structured pruning can lead to greater loss compared to unstructured or semi-structured pruning.

### 2.2. Low-rank pruning

Low-rank pruning applies matrix decomposition techniques, such as Singular Value Decomposition (SVD), to approximate weight matrices with lower-rank representations, thereby reducing both storage and computational demands. This approach, compatible with any GPU, represents large matrices as products of smaller ones, improving computational efficiency. Recent studies (Hsu et al., 2022; Yuan et al., 2023; Wang et al., 2024; Jaiswal et al., 2024; Saha et al., 2024; Kaushal et al., 2023; Sharma et al., 2023; Qinsi et al.) highlight the effectiveness of low-rank decomposition in compressing LLMs. However, despite their flexibility, these methods lag behind semi-structured pruning in performance, often leading to a $2\times$ increase in perplexity (PPL) at the same densities.

## 3. Lossless Low-Rank Compression

### 3.1. Motivation: Information Redundancy in Singular Value Decomposition

For a weight matrix $\mathbf{W} \in \mathbb{R}^{m \times n}$, Low-rank pruning methods (Hsu et al., 2022; Yuan et al., 2023; Wang et al., 2024) decompose the matrix into a product of two low-rank matrices, $\mathbf{W} \approx \mathbf{U}\mathbf{V}^{\mathrm{T}}$, where $\mathbf{U} \in \mathbb{R}^{m \times r}$ and $\mathbf{V}^{\mathrm{T}} \in \mathbb{R}^{r \times n}$, forming a low-rank approximation of $\mathbf{W}$. Consider naive SVD pruning as an example. First, the weight matrix is factorized using SVD as $\mathbf{W} = \mathbf{B}\mathbf{E}\mathbf{A}^{\mathrm{T}}$. Next, the top-$r$ singular values and corresponding singular vectors are retained, denoted as $\mathbf{B}_r$, $\mathbf{E}_r$, and $\mathbf{A}_r^{\mathrm{T}}$. Finally, the singular values $\mathbf{E}_r$ are merged into the singular vectors, yielding $\mathbf{U} = \mathbf{B}_r\mathbf{E}_r$ and $\mathbf{V}^{\mathrm{T}} = \mathbf{A}_r^{\mathrm{T}}$.

The number of parameters in the dense weight matrix is $mn$, whereas the total number of parameters in the low-rank matrices is $r(m + n)$. As shown in Figure 1, SVD-based low-rank representations fail to compress the weight matrix when $r$ exceeds half of the matrix dimensions. However, in low-rank decomposition, the orthogonality constraints among singular vectors reduce the effective degrees of freedom. Specifically, for $\mathbf{U}$ and $\mathbf{V}^{\mathrm{T}}$, there are $\binom{r}{2} = \frac{r(r-1)}{2}$ unique pairs of singular vectors for each matrix, and each pair imposes one linear constraint due to orthogonality. Together, these constraints reduce the total degrees of freedom by $r(r-1)$.

Thus, the actual degrees of freedom in the low-rank representation are $r(m + n) - (r^2 - r)$. This reveals redundancy in low-rank representations and suggests the possibility of encoding the same information with fewer parameters by eliminating it.

> *Question:* Can we design a matrix factorization method that reduces parameters to $r(m+n) - (r^2 - r)$ without losing representational power?

## 3.2. Pivoting Factorization

To address the previously discussed question, we propose **Pi**voting **Fa**ctorization (**PIFA**), a novel matrix factorization method. For any low-rank matrices, which can be obtained by any low-rank pruning methods, PIFA further reduces parameters without inducing additional loss.

The process of Pivoting Factorization is illustrated in Figure 2(b). Given a weight matrix already decomposed into low-rank matrices $\mathbf{U} \in \mathbb{R}^{m \times r}$ and $\mathbf{V}^T \in \mathbb{R}^{r \times n}$, we first multiply $\mathbf{U}$ and $\mathbf{V}^T$ to get the singular matrix, $\mathbf{W}' = \mathbf{U}\mathbf{V}^T$. Since $\mathbf{W}'$ has rank $r$, it contains $r$ linearly independent rows, also referred to as pivot rows. The set of linearly independent rows can be identified using LU or QR decomposition with pivoting (Businger & Golub, 1971). Let $\mathcal{I}$ represent the set of row indices corresponding to the $r$ pivot rows of $\mathbf{W}'$. Thus, any non-pivot row can be expressed as a linear combination of these $r$ pivot rows. Denoting the set of non-pivot row indices as $\mathcal{I}^c = \{1, 2, ..., m\} \backslash \mathcal{I}$, then we define:

$$\mathbf{W}_p = \mathbf{W}'[\mathcal{I}, :], \quad \mathbf{W}_{np} = \mathbf{W}'[\mathcal{I}^c, :] \quad (1)$$

where $\mathbf{W}_p \in \mathbb{R}^{r \times n}$ is the pivot-row matrix, and $\mathbf{W}_{np} \in \mathbb{R}^{(m-r) \times n}$ is the non-pivot-row matrix. With the definition of non-pivot rows, non-pivot-row matrix can be expressed as:

$$\mathbf{W}_{np} = \mathbf{C}\mathbf{W}_p \quad (2)$$

where $\mathbf{C} \in \mathbb{R}^{(m-r) \times r}$ is the coefficient matrix. Algorithm 1 details the PIFA process, which generates the components of a PIFA layer: pivot-row indices $\mathcal{I}$, the pivot-row matrix $\mathbf{W}_p$, and the coefficient matrix $\mathbf{C}$. Algorithm 2 describes the inference procedure for the PIFA layer, which leverages $\mathbf{W}_p$, $\mathbf{C}$ and $\mathcal{I}$ to compute the output.

---

**Algorithm 1** Pivoting Factorization

**input** Low-rank matrix $\mathbf{W}' \in \mathbb{R}^{m \times n}$ with rank $r$
1: Use QR (or LU) decomposition with pivoting to find pivot-row indices: $\mathcal{I} \leftarrow \text{QR}_{\text{pivot}}(\mathbf{W}')$
2: Define $\mathcal{I}^c = \{1, 2, \ldots, m\} \backslash \mathcal{I}$, the complement of $\mathcal{I}$, representing non-pivot row indices
3: Compute pivot-row matrix: $\mathbf{W}_p \leftarrow \mathbf{W}'[\mathcal{I}, :]$
4: Compute non-pivot-row matrix: $\mathbf{W}_{np} \leftarrow \mathbf{W}'[\mathcal{I}^c, :]$
5: Compute coefficient matrix $\mathbf{C}$ by solving matrix equation: $\mathbf{C} \leftarrow \text{linsolve}(\mathbf{W}_{np} = \mathbf{C}\mathbf{W}_p)$
**output** PIFA layer $P$: 1) Pivot-row indices $\mathcal{I} \in \mathbb{R}^r$; 2) pivot-row matrix $\mathbf{W}_p \in \mathbb{R}^{r \times n}$; 3) coefficient matrix $\mathbf{C} \in \mathbb{R}^{(m-r) \times r}$ for non-pivot rows

---

**Algorithm 2** PIFA Layer

**input** Input $\mathbf{X} \in \mathbb{R}^{n \times b}$, where $b$ is batch size; pivot-row indices $\mathcal{I} \in \mathbb{R}^r$; pivot-row matrix $\mathbf{W}_p \in \mathbb{R}^{r \times n}$; coefficient matrix $\mathbf{C} \in \mathbb{R}^{(m-r) \times r}$ for non-pivot rows
1: Define $\mathcal{I}^c = \{1, 2, \ldots, m\} \backslash \mathcal{I}$ representing non-pivot row indices
2: Compute output of pivot channels: $\mathbf{Y}_p \leftarrow \mathbf{W}_p \mathbf{X}$
3: Compute output of non-pivot channels: $\mathbf{Y}_{np} \leftarrow \mathbf{C}\mathbf{Y}_p$
4: Assign pivot channels to output: $\mathbf{Y}[\mathcal{I}, :] \leftarrow \mathbf{Y}_p$
5: Assign non-pivot channels to output: $\mathbf{Y}[\mathcal{I}^c, :] \leftarrow \mathbf{Y}_{np}$
**output** $\mathbf{Y}$

---

## 3.3. Memory and Computational Cost of PIFA

**Memory cost of PIFA.** For each low-rank weight matrix $\mathbf{W}'$, PIFA needs to store $\mathcal{I}$, $\mathbf{W}_p$ and $\mathbf{C}$, totaling $r(m + n) - r^2 + r$. Figure 1 illustrates the relationship between the number of parameters in PIFA, traditional low-rank decomposition, and a dense weight matrix (square).

Since $r(m + n) > r(m + n) - r^2 + r$ for any rank $r$, PIFA consistently requires less memory than traditional low-rank decomposition. For the comparison with dense weight matrix, because $r < \min(m, n)$, we have:

$$(m - r)(n - r) > 0 \quad \Rightarrow \quad mn > r(m + n) - r^2 \quad (3)$$

Neglecting the pivot-row index $\mathcal{I}$, which has negligible memory overhead compared to other variables, PIFA could consistently consumes less memory than a dense weight matrix. In contrast, traditional low-rank decomposition may exceed the memory cost of dense matrices when $r > \frac{mn}{m+n}$.

**Computational cost of PIFA.** We analyze the computational cost of each linear layer for an input batch size $b$, where $\mathbf{X} \in \mathbb{R}^{n \times b}$. We compute the FLOPs for each method as follows:

- For the dense linear layer $\mathbf{Y} = \mathbf{W}\mathbf{X}$, where $\mathbf{W} \in \mathbb{R}^{m \times n}$ and $\mathbf{X} \in \mathbb{R}^{n \times b}$, the computational cost is $2mnb$ FLOPs.
- For the traditional low-rank layer $\mathbf{Y} = \mathbf{U}\mathbf{V}^T\mathbf{X}$, where $\mathbf{U} \in \mathbb{R}^{m \times r}$ and $\mathbf{V}^T \in \mathbb{R}^{r \times n}$, the computational cost includes: $\mathbf{V}^T\mathbf{X}$ ($2rnb$) and $\mathbf{U}(\mathbf{V}^T\mathbf{X})$ ($2mrb$). The total cost is $2rnb + 2mrb = 2br(m + n)$ FLOPs.
- For the PIFA layer (Algorithm 2), the computational cost includes: $\mathbf{Y}_p \leftarrow \mathbf{W}_p\mathbf{X}$ ($2rnb$) and $\mathbf{Y}_{np} \leftarrow \mathbf{C}\mathbf{Y}_p$ $2br(m - r)$. The total cost is $2rnb + 2br(m - r) = 2br(m + n - r)$ FLOPs.

PIFA's computational cost is proportional to its memory cost, differing only by a factor of $2b$. As a result, PIFA consistently outperforms both dense linear layers and traditional low-rank layers in computational efficiency.

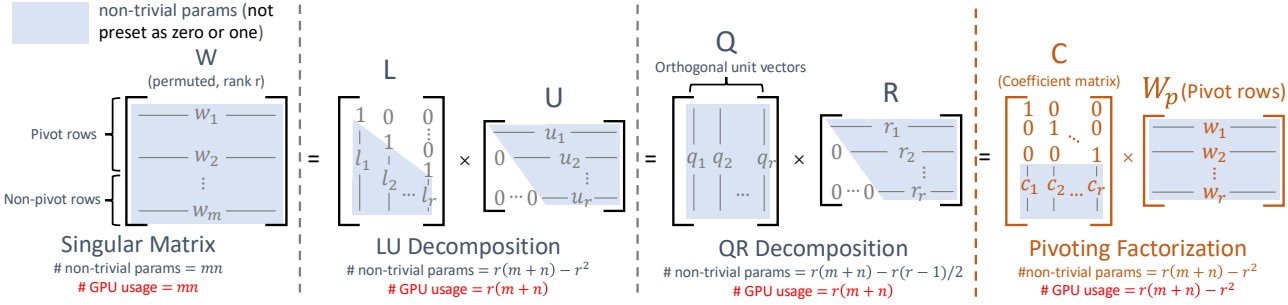

Figure 3: **Pivoting Factorization vs. LU and QR decompositions.** Applied to the permuted matrix (pivot rows at the top), Pivoting Factorization avoids the trapezoidal distribution of non-trivial parameters in LU decomposition, instead reorganizing them into a rectangular pattern. This structure optimizes GPU memory usage and reduces computation overhead.

**Comparison with LU and QR decomposition.** Figure 3 compares the structure of LU and QR decomposition with Pivoting Factorization on a permuted weight matrix, where pivot rows have already been moved to the top. LU decomposition retains the same number of non-trivial parameters (i.e., those not preset as zero or one) as Pivoting Factorization. However, the trapezoidal distribution of non-trivial parameters in LU decomposition complicates efficient storage and computation on the GPU. In contrast, PIFA reorganizes all non-trivial parameters into a rectangular distribution, which is more GPU-friendly for storage and computation. Thus, Pivoting Factorization is more efficient for GPU computation.

# 4. Online Error-Accumulation-Minimization Reconstruction

In addition to PIFA, we propose a novel **Online Error-Accumulation-Minimization Reconstruction (M)** method (illustrated in Figure 2(a)) to reconstruct the low-rank matrix before applying PIFA.

A low-rank pruning step is first required to obtain the low-rank matrices $\mathbf{U}$ and $\mathbf{V}^{\mathrm{T}}$ before reconstruction. To achieve this, we adopt the pruning method from SVD-LLM (Wang et al., 2024), which has demonstrated superior performance among existing methods.

SVD-LLM first introduced low-rank matrix reconstruction. It updates $\mathbf{U}$ using a closed-form least squares solution:

$$\begin{aligned} \mathbf{U}_r &= \arg\min_{\mathbf{U}} \|\mathbf{WX} - \mathbf{UV}^{\mathrm{T}}\mathbf{X}\|_{\mathrm{F}} \\ &= \mathbf{WXD}^{\mathrm{T}}(\mathbf{DD}^{\mathrm{T}})^{-1}, \mathbf{D} = \mathbf{V}^{\mathrm{T}}\mathbf{X} \end{aligned} \quad (4)$$

where $\mathbf{X}$ is the calibration data. We improve Equation 4 in the following aspects:

① **Online algorithm.** Equation 4 requires loading the entire calibration dataset $\mathbf{X}$ into GPU memory to compute the least squares solution. As a result, the number of calibration samples is limited to a maximum of 16 on LLaMA2-7B (4 on LLaMA2-70B) with a 48GB A6000 GPU, leading to overfitting to the calibration data (see Section 5.3).

Applying the associative property of matrix multiplication, we reformulate Equation 4 into its online version:

$$\mathbf{U}_r = \mathbf{W}(\mathbf{X}\mathbf{X}^{\mathrm{T}})\mathbf{V}(\mathbf{V}^{\mathrm{T}}(\mathbf{X}\mathbf{X}^{\mathrm{T}})\mathbf{V})^{-1} \quad (5)$$

The term $\mathbf{X}\mathbf{X}^{\mathrm{T}}$ can be computed incrementally as $\mathbf{X}\mathbf{X}^{\mathrm{T}} = \sum_{i=1}^{b} \mathbf{x}_i\mathbf{x}_i^{\mathrm{T}}$, where $\mathbf{x}_i$ represents the input of sample $i$. As $\mathbf{X}\mathbf{X}^{\mathrm{T}} \in \mathbb{R}^{n \times n}$, the memory consumption of the online least squares solution remains constant, regardless of the number of calibration samples.

② **Error Accumulation Minimization.** Existing reconstruction methods in low-rank pruning (Wang et al., 2024) and semi-structured pruning (Frantar & Alistarh, 2023; Li et al., 2024) rely solely on one data flow, i.e. low-rank data flow in Figure 2. This approach allows accumulated errors from previous modules to propagate through the reconstruction process, potentially degrading performance, as each subsequent module is optimized based on an already-degraded data flow.

Our method mitigates this issue by correcting accumulated error at each module, realigning it with the dense data flow:

$$\min \|\mathbf{W}\mathbf{X}_o - \mathbf{U}\mathbf{V}^{\mathrm{T}}\mathbf{X}_u\|_{\mathrm{F}} \quad (6)$$

where $\mathbf{X}_o$ represents the dense data flow input, produced by the previous layer's dense weight, and $\mathbf{X}_u$ represents the low-rank data flow input, produced by the previous layer's low-rank weight. This ensures that each module's output remains aligned with the output of original model, recovering the accumulated error in $\mathbf{X}_u$.

However, in practical experiments, we observe that relying solely on the dense data flow output $\mathbf{W}\mathbf{X}_o$ as the reconstruction target tends to overfit the reconstructed low-rank weight to the distribution of the calibration data. Using a combination of dense and low rank data flow outputs mitigates

overfitting:

$$\mathbf{Y}_t = \lambda \mathbf{W} \mathbf{X}_o + (1 - \lambda) \mathbf{W} \mathbf{X}_u \tag{7}$$

where $\lambda$ is the **mix ratio**. The optimization target becomes $\min \|\mathbf{Y}_t - \mathbf{U}\mathbf{V}^{\mathrm{T}}\mathbf{X}_u\|_{\mathrm{F}}$. The low-rank data flow output serves as a regularization term, minimizing the distance between $\mathbf{U}\mathbf{V}^{\mathrm{T}}\mathbf{X}_u$ and $\mathbf{W}\mathbf{X}_u$. Since $\mathbf{W}$ has been optimized on a much larger and more diverse pre-training dataset, using it as a constraint helps $\mathbf{U}\mathbf{V}^{\mathrm{T}}$ generalize better and prevents overfitting to the limited calibration data. Empirically, we found that the mix ratio $\lambda = 0.25$ achieves the best performance (see ablation study in Section 5.3).

③ **Reconstructing both $\mathbf{U}$ and $\mathbf{V}^{\mathrm{T}}$.**  Equation 4 reconstructs only $\mathbf{U}$. We find it beneficial to also update $\mathbf{V}^{\mathrm{T}}$ and provide the closed-form solution:

$$\begin{aligned} \mathbf{V}_r^{\mathrm{T}} &= \arg \min_{\mathbf{V}^{\mathrm{T}}} \|\mathbf{Y}_t - \mathbf{U}\mathbf{V}^{\mathrm{T}}\mathbf{X}\|_{\mathrm{F}} \\ &= (\mathbf{U}^{\mathrm{T}}\mathbf{U})^{-1}\mathbf{U}^{\mathrm{T}}\mathbf{Y}_t\mathbf{X}^{\mathrm{T}}(\mathbf{X}\mathbf{X}^{\mathrm{T}})^{-1} \end{aligned} \tag{8}$$

The proof is provided in Appendix A. Updating $\mathbf{V}^{\mathrm{T}}$ can also be performed online by incrementally computing $\mathbf{Y}\mathbf{X}^{\mathrm{T}}$ and $\mathbf{X}\mathbf{X}^{\mathrm{T}}$.

# 5. Experiments

**MPIFA.** We combine Online Error-Accumulation-**M**inimization Reconstruction with **Pi**voting **Fa**ctorization into an end-to-end low-rank compression method, **MPIFA** (illustrated in Figure 2). MPIFA proceeds as follows: ① First, Online Error-Accumulation-Minimization Reconstruction is applied to obtain and refine the low-rank matrices $\mathbf{U}_r$ and $\mathbf{V}_r^{\mathrm{T}}$; ② Then, PIFA decomposes the singular matrix $\mathbf{W}' = \mathbf{U}_r\mathbf{V}_r^{\mathrm{T}}$ into $\mathcal{I}, \mathbf{W}_p, \mathbf{C} \leftarrow \mathrm{PIFA}(\mathbf{W}')$, which are stored in a PIFA layer that replaces the original linear layer.

**MPIFA_{NS}.** Compared to semi-structured sparsity, MPIFA offers the advantage of allowing non-uniform sparsity for each module. We term the **N**on-uniform **S**parsity version of MPIFA as **MPIFA_{NS}**. The detailed implementation of MPIFA_{NS} can be found in Appendix B.2.

**Models and Datasets.** We apply MPIFA to pre-trained LLMs: LLaMA2 (7B, 13B, 70B) (Touvron et al., 2023b) and LLaMA3 (8B) (Dubey et al., 2024). Both the calibration data and perplexity (PPL) evaluations are based on the WikiText2 dataset (Merity et al., 2022), with a sequence length of 2048 tokens for all experiments.

## 5.1. Main Result

**Comparison with other low-rank pruning.** We evaluate MPIFA against state-of-the-art low-rank pruning methods:

ASVD (Yuan et al., 2023) and SVD-LLM (Wang et al., 2024). Vanilla SVD is also included for reference. SVD-LLM has two versions, as detailed in their original paper. Following their approach, we select the best-performing version for each density. Results for both versions of SVD-LLM are provided in Table 5. MPIFA utilizes 128 calibration samples and sets $\lambda = 0.25$. For all models except LLaMA-2-70B, MPIFA reconstructs both $\mathbf{U}$ and $\mathbf{V}^{\mathrm{T}}$. For LLaMA-2-70B, MPIFA reconstructs only $\mathbf{U}$.

Table 2 displays the test perplexity (PPL) of each low-rank pruning method on WikiText2 under 0.4-0.9 density, which is defined as the proportion of parameters remaining compared with the original model. The results show that MPIFA significantly outperforms other low-rank pruning method, reducing the perplexity gap by **66.4%** (LLaMA2-7B), **53.8%** (LLaMA2-13B), **40.7%** (LLaMA2-70B), and **72.7%** (LLaMA3-8B) on average. The perplexity gap is defined as the difference between the perplexity of a low-rank pruning method and the original model.

**Comparison with semi-structured pruning.** We further compare MPIFA with 2:4 semi-structured pruning methods: magnitude pruning (Zhu & Gupta, 2017), and two recent state-of-the-art works Wanda (Sun et al., 2024), and RIA (Zhang et al., 2024). According to (Mishra et al., 2021), for 16-bit operands, 2:4 sparse leads to $\sim$44% savings in GPU memory. Therefore, we compare 2:4 sparse method with MPIFA at 0.55 density to ensure that all methods achieve the same memory reduction (see Table 6 for memory comparison).

Table 3 shows that MPIFA_{NS} outperforms 2:4 pruning methods, reducing the perplexity gap by **21.7%** for LLaMA2-7B and **3.2%** for LLaMA2-13B.

**Fine-tuning After Pruning.** We investigate how fine-tuning helps recover the performance loss caused by low-rank pruning. The detailed experimental settings are provided in Appendix B.3. As shown in Table 4, fine-tuned MPIFA_{NS} achieves the best performance among all fine-tuned pruning methods, bringing its perplexity close to the dense baseline at 55% density. Unlike semi-structured methods, which cannot accelerate the backward pass due to transposed weight tensors violating the 2:4 constraint (Mishra et al., 2021), PIFA and other low-rank methods enable acceleration in both the forward and backward passes.

## 5.2. Inference Speedup and Memory Reduction

**PIFA layer vs low-rank layer.** The PIFA layer achieves significant savings in both memory and computation time, as shown in Figure 7, which compares its actual runtime and memory usage with those of a standard linear layer and an SVD-based low-rank layer. In FP32, at 50% density,

Table 2: **Perplexity (↓) at different parameter density** (proportion of remaining parameters relative to the original model) on WikiText2. The best-performing method is highlighted in **bold**.

| Model | Method | Density | | | | | | |
|---|---|---|---|---|---|---|---|---|
| | | 100% | 90% | 80% | 70% | 60% | 50% | 40% |
| LLaMA2-7B | SVD | 5.47 | 16063 | 18236 | 30588 | 39632 | 53179 | 65072 |
| | ASVD | | 5.91 | 9.53 | 221.6 | 5401 | 26040 | 24178 |
| | SVD-LLM | | 7.27 | 8.38 | 10.66 | 16.11 | 27.19 | 54.20 |
| | MPIFA | | **5.69** | **6.16** | **7.05** | **8.81** | **12.77** | **21.25** |
| LLaMA2-13B | SVD | 4.88 | 2168 | 6177 | 37827 | 24149 | 14349 | 41758 |
| | ASVD | | 5.12 | 6.67 | 17.03 | 587.1 | 3103 | 4197 |
| | SVD-LLM | | 5.94 | 6.66 | 8.00 | 10.79 | 18.38 | 42.79 |
| | MPIFA | | **5.03** | **5.39** | **7.12** | **7.41** | **10.30** | **16.72** |
| LLaMA2-70B | SVD | 3.32 | 6.77 | 17.70 | 203.7 | 2218 | 6803 | 15856 |
| | ASVD | | OOM | OOM | OOM | OOM | OOM | OOM |
| | SVD-LLM | | 4.12 | 4.58 | 5.31 | 6.60 | 9.09 | 14.82 |
| | MPIFA | | **3.54** | **3.96** | **4.58** | **5.54** | **7.40** | **12.01** |
| LLaMA3-8B | SVD | 6.14 | 463461 | 626967 | 154679 | 62640 | 144064 | 216552 |
| | ASVD | | 9.37 | 275.6 | 12553 | 21756 | 185265 | 13504 |
| | SVD-LLM | | 9.83 | 13.62 | 23.66 | 42.60 | 83.46 | 163.5 |
| | MPIFA | | **6.93** | **8.31** | **10.83** | **16.41** | **28.90** | **47.02** |

Table 3: **Perplexity (↓) comparison with semi-structured pruning** under the same memory reduction on WikiText2. The best performance pruning method is indicated in **bold**. MPIFA$_{NS}$ means MPIFA using non-uniform sparsity.

| Method | LLaMA2-7B | LLaMA2-13B |
|---|---|---|
| Dense | 5.47 | 4.88 |
| Magnitude 2:4 | 37.77 | 8.89 |
| Wanda 2:4 | 11.40 | 8.33 |
| RIA 2:4 | 10.85 | 8.03 |
| SVD 55% | 69128 | 24947 |
| ASVD 55% | 9370 | 2039 |
| SVD-LLM 55% | 20.43 | 13.69 |
| MPIFA$_{NS}$ 55% | **9.68** | **7.93** |

Table 4: **Perplexity (↓) of pruned models after fine-tuning** on WikiText2 (LLaMA2-7B). The best-performance pruning method is indicated in **bold**.

| Method | LLaMA2-7B |
|---|---|
| Dense | 5.47 |
| Magnitude 2:4 | 6.63 |
| Wanda 2:4 | 6.40 |
| RIA 2:4 | 6.37 |
| SVD 55% | 9.24 |
| ASVD 55% | 8.64 |
| SVD-LLM 55% | 7.36 |
| MPIFA$_{NS}$ 55% | **6.34** |

PIFA achieves $47.6\%$ memory savings and 1.95x speedup, closely matching the ideal memory and speedup. Additionally, at the same rank, PIFA consistently achieves higher compression and faster inference than the low-rank layer. For example, at $r/d = 0.5$, PIFA losslessly compresses the memory of the low-rank layer by $24.2\%$ and reduces inference time by $24.6\%$.

**PIFA layer vs semi-sparse layer.** Figure 4 and Table 6 compare the speedup and memory usage of the PIFA layer with 2:4 semi-sparse linear layers. The semi-sparse lay-

ers are implemented using cuSPARSELt[2] or CUTLASS[3] library, with the reported speedup representing the higher value between the two implementations. The results span various dimensions on A6000 and A100 GPUs. PIFA demonstrates consistently superior efficiency, achieving the highest speedup and lowest memory usage in all configurations except $d = 4096$. Notably, PIFA's acceleration increases with matrix dimensions, reflecting its scalability and computational effectiveness. As shown in Table 6, at $d = 32768$, PIFA with 55% density achieves a **2.1×** speedup, while 2:4 (CUTLASS) is slower than the dense

[2]https://docs.nvidia.com/cuda/cusparselt/
[3]https://github.com/NVIDIA/cutlass

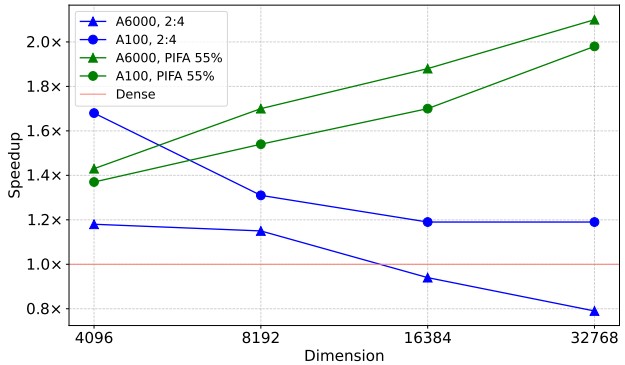

Figure 4: **Layerwise speedup of the PIFA layer and semi-sparse layers** across various dimensions, compared to dense linear layers on the same GPU, with a sequence length of 2048 and a batch size of 32, using FP16 (FP32 is not supported by 2:4 sparsity in `torch.sparse`). PIFA shows increasing speedup as the dimension grows. Detailed values are provided in Table 6.

linear layer, and 2:4 (cuSPARSELt) raises an error.

**End-to-end LLM inference.** Table 7 compares the end-to-end inference throughput and memory usage of MPIFA$_{NS}$ with semi-sparsity (2:4 cuSPARSELt and CUTLASS) on LLaMA2-7B and LLaMA2-13B models in FP16. MPIFA$_{NS}$ consistently outperforms semi-sparsity in both throughput and memory efficiency at $55\%$ density. Furthermore, the operations supported by semi-sparsity are limited in `torch.sparse`, resulting in errors when the KV cache is enabled, which further limits the application of semi-sparse for LLM inference.

### 5.3. Ablation Study

**Impact of PIFA and M** Table 5 presents an ablation study that evaluates the impact of our Online Error-Accumulation-Minimization Reconstruction (denoted as M) and Pivoting Factorization (PIFA) on perplexity across varying parameter densities. The methods compared include:

- **W:** Using only the pruning step of SVD-LLM (truncation-aware data whitening).
- **W + U:** Applying SVD-LLM's pruning followed by full-batch reconstruction.
- **W + M:** Employing our Online Error-Accumulation-Minimization Reconstruction, which incorporates SVD-LLM's pruning as the initial step.
- **W + M + PIFA:** Combining Online Error-Accumulation-Minimization Reconstruction with PIFA (denoted as MPIFA).

The results reveal several key findings:

1. **Full-batch reconstruction (W + U) occasionally**

**worsens perplexity compared to using only the pruning step (W).** This highlights the drawbacks of full-batch methods, as overfitting to the limited calibration data can degrade performance.

2. **Our reconstruction method (W + M) consistently outperforms full-batch reconstruction (W + U) and pruning alone (W) across all models and densities.** This demonstrates the effectiveness of Online Error-Accumulation-Minimization Reconstruction in reducing error accumulation and improving the compression of low-rank matrices.

3. **PIFA further improves performance when combined with M.** The W + M + PIFA configuration achieves the best perplexity across all settings, validating the advantage of applying PIFA for additional parameter reduction without inducing any additional loss.

These findings emphasize the significance of M and PIFA in achieving superior low-rank pruning performance.

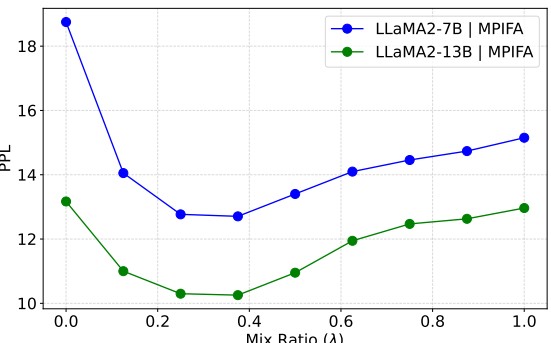

Figure 5: **Impact of mix ratio.** With 0.5 density, MPIFA achieves lowest PPL when the mix ratio $\lambda$ in Equation 7 is around 0.25.

**Impact of mix ratio $\lambda$ in M.** The mix ratio $\lambda$ in Equation 7 determines the proportion of the dense data flow in the reconstruction target. As shown in Figure 5, using a moderate ratio $\lambda = 0.25$, MPIFA achieves significantly lower PPL compared to $\lambda = 0$, where the reconstruction target relies solely on the low-rank data flow, as in previous studies (Wang et al., 2024; Frantar & Alistarh, 2023) did. This demonstrates the effectiveness of our error-accumulation-corrected strategy, in which the dense data flow output is beneficial as part of the reconstruction target. In Figure 5, we also observe that an excessively large $\lambda$ increases PPL, indicating overfitting to the calibration data.

**Impact of Calibration Sample Size.** M depends on calibration data to accurately estimate $\mathbf{U}$ and $\mathbf{V}^{\mathrm{T}}$. As shown in Figure 6, the perplexity of MPIFA decreases as the number of calibration samples increases. We hypothesize that

Table 5: **Ablation: Impact of PIFA and M on perplexity ($\downarrow$) across parameter densities.** on WikiText2. W denotes using SVD-LLM's pruning only; W + U denotes using SVD-LLM's pruning and full-batch reconstruction; W + M denotes using our Online Error-Accumulation-Minimization Reconstruction, which incorporates SVD-LLM's pruning as the initial step; W + M + PIFA denotes using Online Error-Accumulation-Minimization Reconstruction followed by PIFA, i.e., MPIFA.

| Model | Method | 100% | 90% | 80% | 70% | 60% | 50% | 40% |
|---|---|---|---|---|---|---|---|---|
| | | | | | Density | | | |
| LLaMA2-7B | W | 5.47 | 7.27 | 8.38 | 10.66 | 16.14 | 33.27 | 89.98 |
| | W + U | | 7.60 | 8.84 | 11.15 | 16.11 | 27.19 | 54.20 |
| | W + M | | 6.71 | 7.50 | 8.86 | 11.45 | 16.55 | 25.26 |
| | W + M + PIFA (MPIFA) | | **5.69** | **6.16** | **7.05** | **8.81** | **12.77** | **21.25** |
| LLaMA2-13B | W | 4.88 | 5.94 | 6.66 | 8.00 | 10.79 | 18.38 | 43.92 |
| | W + U | | 6.45 | 7.37 | 9.07 | 12.52 | 20.95 | 42.79 |
| | W + M | | 5.80 | 6.41 | 7.42 | 9.31 | 13.09 | 19.93 |
| | W + M + PIFA (MPIFA) | | **5.03** | **5.39** | **7.12** | **7.41** | **10.30** | **16.72** |
| LLaMA2-70B | W | 3.32 | 4.12 | 4.58 | 5.31 | 6.60 | 9.09 | 14.82 |
| | W + U | | 8.23 | 8.33 | 8.66 | 10.02 | 13.41 | 22.39 |
| | W + M | | 4.15 | 4.63 | 5.31 | 6.46 | 8.72 | 14.11 |
| | W + M + PIFA (MPIFA) | | **3.54** | **3.96** | **4.58** | **5.54** | **7.40** | **12.01** |
| LLaMA3-8B | W | 6.14 | 9.83 | 13.62 | 25.43 | 76.86 | 290.3 | 676.7 |
| | W + U | | 10.63 | 14.66 | 23.66 | 42.60 | 83.46 | 163.5 |
| | W + M | | 9.16 | 11.25 | 15.27 | 23.55 | 36.14 | 53.85 |
| | W + M + PIFA (MPIFA) | | **6.93** | **8.31** | **10.83** | **16.41** | **28.90** | **47.02** |

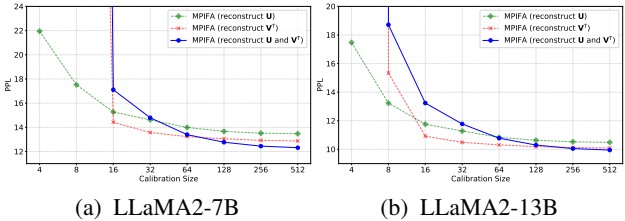

(a) LLaMA2-7B  (b) LLaMA2-13B

Figure 6: **Impact of calibration sample size.** On MPIFA with 0.5 density, reconstructing both $\mathbf{U}$ and $\mathbf{V}^{\mathrm{T}}$ is more sensitive to the number of calibration samples than reconstructing only $\mathbf{U}$.

increasing the number of calibration samples reduces the condition number of the least squares solution, improving numerical stability.

To investigate this, we calculate the condition numbers of $\mathbf{V}^{\mathrm{T}}\mathbf{X}\mathbf{X}^{\mathrm{T}}\mathbf{V}$ in Equation 5 and $\mathbf{X}\mathbf{X}^{\mathrm{T}}$ in Equation 8, as their inverses are required for reconstructing $\mathbf{U}$ and $\mathbf{V}^{\mathrm{T}}$. Figure 8 presents the condition numbers for these matrices in the first layer of LLaMA2-7B. The observed reduction in condition number indicates that the matrices become less singular as the calibration sample size increases, thereby improving numerical stability when solving the least squares equations. This increased stability ultimately results in lower perplexity in the reconstructed model.

**Impact of reconstructing $\mathbf{U}$ and $\mathbf{V}^{\mathrm{T}}$.** Figures 6a and 6b compare the effects of reconstructing only $\mathbf{U}$, only $\mathbf{V}^{\mathrm{T}}$, and both $\mathbf{U}$ and $\mathbf{V}^{\mathrm{T}}$ across different calibration sizes. The results indicate that with sufficient calibration samples, reconstructing both $\mathbf{U}$ and $\mathbf{V}^{\mathrm{T}}$ achieves lower perplexity than reconstructing only $\mathbf{U}$ or only $\mathbf{V}^{\mathrm{T}}$.

# 6. Conclusion and Discussion

In this work, we proposed **MPIFA**, an end-to-end, retraining-free low-rank pruning framework that integrates **Pivoting Factorization (PIFA)** and an **Online Error-Accumulation-Minimization Reconstruction (M)** algorithm. PIFA serves as a meta low-rank representation that further compresses existing low-rank decompositions, achieving superior memory savings and inference speedup without additional loss. While this allows PIFA to integrate seamlessly with various low-rank compression techniques, it does not reduce matrix rank on its own and must be applied alongside a low-rank compression method. Meanwhile, M mitigates error accumulation, leading to improved performance. Together, these innovations enable MPIFA to achieve performance comparable to semi-structured pruning while surpassing it in GPU acceleration and compatibility. Future work could explore integrating PIFA into the pretraining stage, as PIFA is fully differentiable, enabling potential efficiency gains during model training.

## Acknowledgements

This work was supported by the Zhou Yahui Chair Professorship award of Tsinghua University (to CVC), the National High-Level Talent Program of the Ministry of Science and Technology of China (grant number 20241710001, to CVC).

## Impact Statement

This paper presents work whose goal is to advance the field of Machine Learning. There are many potential societal consequences of our work, none which we feel must be specifically highlighted here.

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

# Appendix

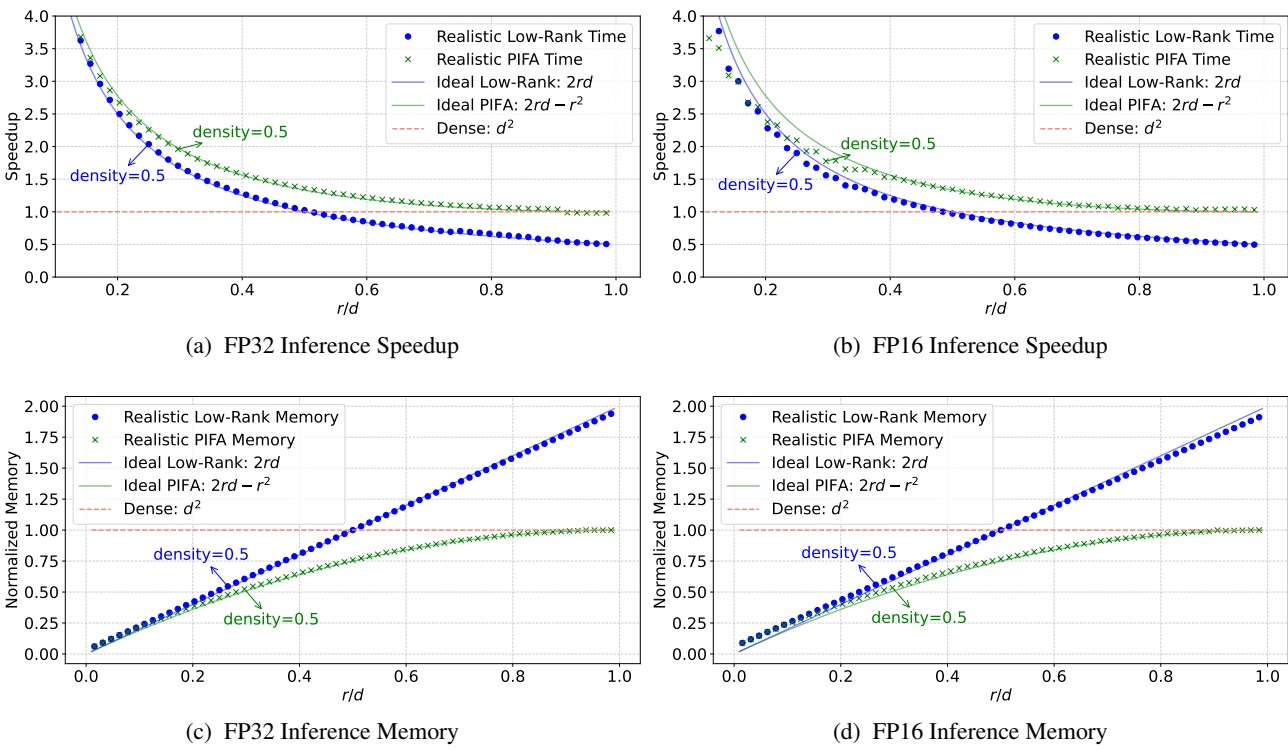

(a)  FP32 Inference Speedup

(b)  FP16 Inference Speedup

(c)  FP32 Inference Memory

(d)  FP16 Inference Memory

Figure 7: **Efficiency of PIFA layer** under various ranks, with sequence length = 2048, batch size = 32, and dimension = 8192 on FP32 and FP16 on A6000 GPU. At $50\%$ density, PIFA achieves $47.6\%$ memory savings and $1.95\times$ speedup on FP32. These results guarantee that the overhead of both time and memory of the PIFA layer is quite low.

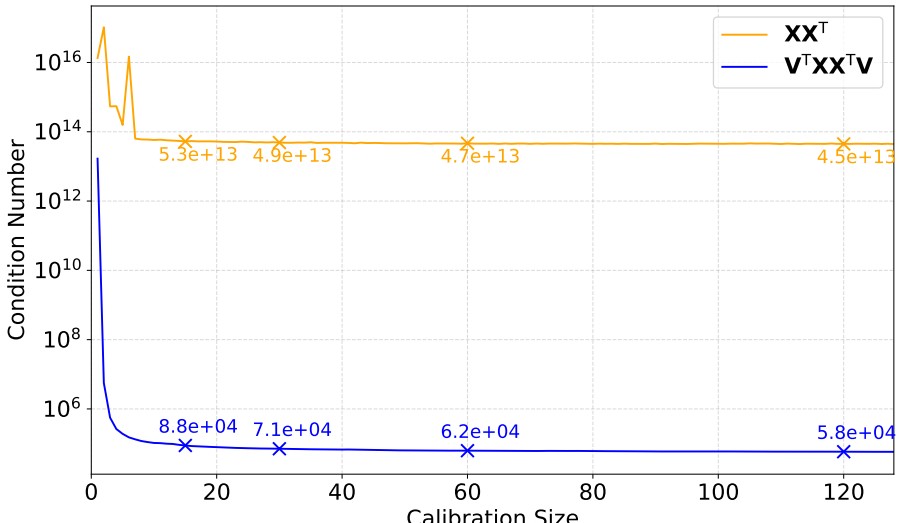

Figure 8: **Condition number.** Condition numbers of $\mathbf{V}^{\mathrm{T}}\mathbf{X}\mathbf{X}^{\mathrm{T}}\mathbf{V}$ (Equation 5) and $\mathbf{X}\mathbf{X}^{\mathrm{T}}$ (Equation 8) for the first layer of LLaMA2-7B, whose inverses are used in reconstructing $\mathbf{U}$ and $\mathbf{V}^{\mathrm{T}}$. Larger calibration sizes reduce condition numbers, enhancing numerical stability and lowering perplexity.

Table 6: **Efficiency of PIFA layer and semi-sparse layer** across different dimensions, compared to dense linear at same dimension on same GPU, with sequence length of 2048 and batch size of 32, using FP16 (FP32 is not supported by 2:4 sparsity in `torch.sparse`). The highest speedup and lowest memory are indicated in **bold**. PIFA shows increasing speedup as the dimension grows. [†]For matrix multiplication with weight matrix shape 32768×32768, cuSPARSELt raises CUDA error.

| | | | Dimension | | | |
|---|---|---|---|---|---|---|
| | GPU | Kernel | 32768 | 16384 | 8192 | 4096 |
| Speedup | A6000 | 2:4 (cuSPARSELt) | Error[†] | 0.94× | 0.97× | 1.09× |
| | | 2:4 (CUTLASS) | 0.79× | 0.92× | 1.15× | 1.18× |
| | | PIFA 55% | **2.10×** | **1.88×** | **1.70×** | **1.43×** |
| | A100 | 2:4 (cuSPARSELt) | Error[†] | 1.19× | 1.31× | **1.68×** |
| | | 2:4 (CUTLASS) | 1.19× | 1.12× | 1.09× | 1.52× |
| | | PIFA 55% | **1.98×** | **1.70×** | **1.54×** | 1.37× |
| Memory | | 2:4 (cuSPARSELt / CUTLASS) | 0.564 | 0.569 | 0.589 | 0.651 |
| | | PIFA 55% | **0.552** | **0.558** | **0.578** | **0.645** |

Table 7: **End-to-end efficiency of MPIFA$_{NS}$** on LLaMA2 models, on FP16 (FP32 isn't supported by semi-sparse). The highest throughput and lowest memory are indicated in **bold**. MPIFA$_{NS}$ consistently outperforms semi-sparse in both throughput and memory with 55% density. [†]Enabling KV cache for semi-sparse model will cause `SparseSemiStructuredTensorCUSPARSELT only supports a specific set of operations, can't perform requested op (expand.default)`

| Model | Metrics | GPU | Use KV Cache | Dense | 2:4 (cuSPARSELt) | 2:4 (CUTLASS) | MPIFA$_{NS}$ 55% |
|---|---|---|---|---|---|---|---|
| llama2-7b | Throughput (token/s) | A6000 | No | 354.9 | 306.6 | 327.5 | **472.6** |
| | | | Yes | 3409 | Error[†] | Error | **4840** |
| | | A100 | No | 614.8 | 636.2 | 582.3 | **822.2** |
| | | | Yes | 6918 | Error | Error | **7324** |
| | Memory (GB) | | | 12.55 | 7.274 | 7.290 | **7.174** |
| llama2-13b | Throughput (token/s) | A6000 | No | 190.0 | 163.2 | 180.0 | **268.7** |
| | | | Yes | 2156 | Error | Error | **2721** |
| | | A100 | No | 345.4 | 362.4 | 321.5 | **473.3** |
| | | | Yes | 4217 | Error | Error | **4532** |
| | Memory (GB) | | | 24.36 | 13.90 | 13.99 | **13.69** |

# A. Closed-Form Solution of $\mathbf{V}^\mathrm{T}$

We aim to prove that minimizing the Frobenius norm $\|\mathbf{Y} - \mathbf{U}\mathbf{V}^\mathrm{T}\mathbf{X}\|_F$ with respect to $\mathbf{V}^\mathrm{T}$ is equivalent to performing the following two-step optimization:

1. First, minimize $\|\mathbf{Y} - \mathbf{W}\mathbf{X}\|_F$ with respect to $\mathbf{W}$.

2. Then, minimize $\|\mathbf{W} - \mathbf{U}\mathbf{V}^\mathrm{T}\|_F$ with respect to $\mathbf{V}^\mathrm{T}$.

We begin by directly minimizing $\|\mathbf{Y} - \mathbf{U}\mathbf{V}^\mathrm{T}\mathbf{X}\|_F^2$ with respect to $\mathbf{V}^\mathrm{T}$.

## A.1. Direct Optimization

$$
\begin{aligned}
f(\mathbf{V}) &= \|\mathbf{Y} - \mathbf{U}\mathbf{V}^\mathrm{T}\mathbf{X}\|_F^2 \\
&= \mathrm{Tr}\left((\mathbf{Y} - \mathbf{U}\mathbf{V}^\mathrm{T}\mathbf{X})^\mathrm{T}(\mathbf{Y} - \mathbf{U}\mathbf{V}^\mathrm{T}\mathbf{X})\right) \\
&= \mathrm{Tr}\left(\mathbf{Y}^\mathrm{T}\mathbf{Y} - \mathbf{Y}^\mathrm{T}\mathbf{U}\mathbf{V}^\mathrm{T}\mathbf{X} - \mathbf{X}^\mathrm{T}\mathbf{V}\mathbf{U}^\mathrm{T}\mathbf{Y} + \mathbf{X}^\mathrm{T}\mathbf{V}\mathbf{U}^\mathrm{T}\mathbf{U}\mathbf{V}^\mathrm{T}\mathbf{X}\right) \\
&= \mathrm{Tr}(\mathbf{Y}^\mathrm{T}\mathbf{Y}) - 2\,\mathrm{Tr}(\mathbf{V}^\mathrm{T}\mathbf{X}\mathbf{Y}^\mathrm{T}\mathbf{U}) + \mathrm{Tr}(\mathbf{V}^\mathrm{T}\mathbf{X}\mathbf{X}^\mathrm{T}\mathbf{V}\mathbf{U}^\mathrm{T}\mathbf{U})
\end{aligned}
$$

Let us define intermediate matrices:

$$
\begin{aligned}
\mathbf{A} &= \mathbf{X}\mathbf{Y}^\mathrm{T}\mathbf{U} \\
\mathbf{B} &= \mathbf{X}\mathbf{X}^\mathrm{T} \\
\mathbf{C} &= \mathbf{U}^\mathrm{T}\mathbf{U}
\end{aligned}
$$

The objective function becomes:

$$
f(\mathbf{V}) = \mathrm{const} - 2\,\mathrm{Tr}(\mathbf{V}^\mathrm{T}\mathbf{A}) + \mathrm{Tr}(\mathbf{V}^\mathrm{T}\mathbf{B}\mathbf{V}\mathbf{C})
$$

where "const" denotes terms independent of $\mathbf{V}$.

Compute the gradient of $f(\mathbf{V})$ with respect to $\mathbf{V}$:

$$
\frac{\partial f}{\partial \mathbf{V}} = -2\mathbf{A} + 2\mathbf{B}\mathbf{V}\mathbf{C}
$$

Set the gradient to zero:

$$
-2\mathbf{A} + 2\mathbf{B}\mathbf{V}\mathbf{C} = 0 \implies \mathbf{B}\mathbf{V}\mathbf{C} = \mathbf{A}
$$

Assuming $\mathbf{B}$ and $\mathbf{C}$ are invertible, we solve for $\mathbf{V}$:

$$
\mathbf{V} = \mathbf{B}^{-1}\mathbf{A}\mathbf{C}^{-1}
$$

Substituting back the definitions of $\mathbf{A}, \mathbf{B}, \mathbf{C}$:

$$
\mathbf{V} = (\mathbf{X}\mathbf{X}^\mathrm{T})^{-1}\left(\mathbf{X}\mathbf{Y}^\mathrm{T}\mathbf{U}\right)(\mathbf{U}^\mathrm{T}\mathbf{U})^{-1}
$$

Simplify:

$$
\mathbf{V}^\mathrm{T} = (\mathbf{U}^\mathrm{T}\mathbf{U})^{-1}\mathbf{U}^\mathrm{T}\mathbf{Y}\mathbf{X}^\mathrm{T}(\mathbf{X}\mathbf{X}^\mathrm{T})^{-1}
$$

### A.2. Two-Step Optimization

Now, we perform the two-step optimization and show it leads to the same result.

**First**, minimize $\|\mathbf{Y} - \mathbf{W}\mathbf{X}\|_F^2$ with respect to $\mathbf{W}$.

Compute the gradient:

$$\frac{\partial}{\partial \mathbf{W}}\|\mathbf{Y} - \mathbf{W}\mathbf{X}\|_F^2 = -2(\mathbf{Y} - \mathbf{W}\mathbf{X})\mathbf{X}^{\mathrm{T}}$$

Set the gradient to zero:

$$(\mathbf{Y} - \mathbf{W}\mathbf{X})\mathbf{X}^{\mathrm{T}} = 0 \implies \mathbf{Y}\mathbf{X}^{\mathrm{T}} = \mathbf{W}\mathbf{X}\mathbf{X}^{\mathrm{T}}$$

Assuming $\mathbf{X}\mathbf{X}^{\mathrm{T}}$ is invertible:

$$\mathbf{W} = \mathbf{Y}\mathbf{X}^{\mathrm{T}}(\mathbf{X}\mathbf{X}^{\mathrm{T}})^{-1}$$

**Next**, minimize $\|\mathbf{W} - \mathbf{U}\mathbf{V}^{\mathrm{T}}\|_F^2$ with respect to $\mathbf{V}^{\mathrm{T}}$.

Compute the gradient:

$$\frac{\partial}{\partial \mathbf{V}}\|\mathbf{W} - \mathbf{U}\mathbf{V}^{\mathrm{T}}\|_F^2 = -2\mathbf{U}^{\mathrm{T}}(\mathbf{W} - \mathbf{U}\mathbf{V}^{\mathrm{T}})$$

Set the gradient to zero:

$$\mathbf{U}^{\mathrm{T}}\mathbf{W} = \mathbf{U}^{\mathrm{T}}\mathbf{U}\mathbf{V}^{\mathrm{T}}$$

Assuming $\mathbf{U}^{\mathrm{T}}\mathbf{U}$ is invertible:

$$\mathbf{V}^{\mathrm{T}} = (\mathbf{U}^{\mathrm{T}}\mathbf{U})^{-1}\mathbf{U}^{\mathrm{T}}\mathbf{W}$$

Substitute $\mathbf{W}$:

$$\mathbf{V}^{\mathrm{T}} = (\mathbf{U}^{\mathrm{T}}\mathbf{U})^{-1}\mathbf{U}^{\mathrm{T}}\left(\mathbf{Y}\mathbf{X}^{\mathrm{T}}(\mathbf{X}\mathbf{X}^{\mathrm{T}})^{-1}\right)$$

Simplify:

$$\mathbf{V}^{\mathrm{T}} = (\mathbf{U}^{\mathrm{T}}\mathbf{U})^{-1}\mathbf{U}^{\mathrm{T}}\mathbf{Y}\mathbf{X}^{\mathrm{T}}(\mathbf{X}\mathbf{X}^{\mathrm{T}})^{-1}$$

### A.3. Conclusion

The solution for $\mathbf{V}^{\mathrm{T}}$ obtained through both the direct optimization and the two-step optimization is:

$$\mathbf{V}^{\mathrm{T}} = (\mathbf{U}^{\mathrm{T}}\mathbf{U})^{-1}\mathbf{U}^{\mathrm{T}}\mathbf{Y}\mathbf{X}^{\mathrm{T}}(\mathbf{X}\mathbf{X}^{\mathrm{T}})^{-1}$$

Therefore, minimizing $\|\mathbf{Y} - \mathbf{U}\mathbf{V}^{\mathrm{T}}\mathbf{X}\|_F$ with respect to $\mathbf{V}^{\mathrm{T}}$ is equivalent to first optimizing $\|\mathbf{Y} - \mathbf{W}\mathbf{X}\|_F$ with respect to $\mathbf{W}$, and then optimizing $\|\mathbf{W} - \mathbf{U}\mathbf{V}^{\mathrm{T}}\|_F$ with respect to $\mathbf{V}^{\mathrm{T}}$.

# B. Experiment Details

## B.1. MPIFA

The full method of MPIFA is outlined in Algorithm 3.

---

**Algorithm 3** MPIFA

---

**input** Original weight matrix $\mathbf{W} \in \mathbb{R}^{m \times n}$; calibration input from dense $\mathbf{X}_o \in \mathbb{R}^{n \times b}$; calibration input from low rank $\mathbf{X}_u \in \mathbb{R}^{n \times b}$; target rank $r$; original output ratio $\lambda$

    **Part 1: Online Error-Accumulation-Minimization Reconstruction**
1: Compute dense output as dense input of next module: $\mathbf{Y}_o = \mathbf{W}\mathbf{X}_o$
2: Use SVD-LLM's pruning (truncation-aware data whitening) to convert to low-rank matrix: $\mathbf{U}, \mathbf{V}^{\mathrm{T}} \leftarrow \text{SVD-LLM}(\mathbf{W})$
3: Compute $\mathbf{X}\mathbf{X}^{\mathrm{T}}$ accumulatively: $\mathbf{X}\mathbf{X}^{\mathrm{T}} \leftarrow \sum_{i=1}^{b} \mathbf{x}_u^i {\mathbf{x}_u^i}^{\mathrm{T}}$
4: Compute $\mathbf{Y}_t\mathbf{X}^{\mathrm{T}}$ accumulatively: $\mathbf{Y}_t\mathbf{X}^{\mathrm{T}} \leftarrow \sum_{i=1}^{b} (\lambda\mathbf{W}\mathbf{x}_o^i + (1-\lambda)\mathbf{W}\mathbf{x}_u^i){\mathbf{x}_u^i}^{\mathrm{T}}$
5: Reconstruct $\mathbf{U}$ as $\mathbf{U}_r$: $\mathbf{U}_r \leftarrow (\mathbf{Y}_t\mathbf{X}^{\mathrm{T}})\mathbf{V}(\mathbf{V}^{\mathrm{T}}(\mathbf{X}\mathbf{X}^{\mathrm{T}})\mathbf{V})^{-1}$
6: Reconstruct $\mathbf{V}$ as $\mathbf{V}_r$: $\mathbf{V}_r^{\mathrm{T}} \leftarrow (\mathbf{U}_r^{\mathrm{T}}\mathbf{U}_r)^{-1}\mathbf{U}_r^{\mathrm{T}}(\mathbf{Y}_t\mathbf{X}^{\mathrm{T}})(\mathbf{X}\mathbf{X}^{\mathrm{T}})^{-1}$
7: Compute low-rank output as low-rank input of next module: $\mathbf{Y}_u = \mathbf{W}\mathbf{X}_u$

    **Part 2: PIFA**
8: Compute low-rank matrix $\mathbf{W}'$: $\mathbf{W}' \leftarrow \mathbf{U}_r\mathbf{V}_r^{\mathrm{T}}$
9: Use Algorithm 1 to build the PIFA layer $P$ using low-rank matrix $\mathbf{W}'$

**output** PIFA layer $P$

---

A potential issue is that $\mathbf{X}\mathbf{X}^{\mathrm{T}}$ can be singular in some cases, leading to NaN values when calculating the inverse matrix during $\mathbf{V}$ reconstruction. To address this, we leverage prior knowledge that $\mathbf{U}\mathbf{V}^{\mathrm{T}}$ should approximate $\mathbf{W}$ by adding a regularization term to the original optimization target, modifying Equation 8 as follows:

$$
\begin{aligned}
\mathbf{V}_r^{\mathrm{T}} &= \arg\min_{\mathbf{V}^{\mathrm{T}}} \|\mathbf{Y}_t - \mathbf{U}\mathbf{V}^{\mathrm{T}}\mathbf{X}\|_{\mathrm{F}} + \alpha\|\mathbf{W} - \mathbf{U}\mathbf{V}^{\mathrm{T}}\|_{\mathrm{F}} \\
&= (\mathbf{U}^{\mathrm{T}}\mathbf{U})^{-1}\mathbf{U}^{\mathrm{T}}(\mathbf{Y}_t\mathbf{X}^{\mathrm{T}} + \alpha\mathbf{W})(\mathbf{X}\mathbf{X}^{\mathrm{T}} + \alpha\mathbf{I})^{-1}
\end{aligned}
\tag{9}
$$

where $\alpha$ is the regularization coefficient, set to 0.001 in all experiments. Regularization is unnecessary for reconstructing $\mathbf{U}$, as no singularity issues were observed for $\mathbf{V}^{\mathrm{T}}(\mathbf{X}\mathbf{X}^{\mathrm{T}})\mathbf{V}$.

## B.2. MPIFA$_{\mathrm{NS}}$

MPIFA$_{\mathrm{NS}}$ is the non-uniform sparsity variant of MPIFA, designed to leverage different sparsity distributions across model layers and module types for improved performance. It employs 512 calibration samples. This approach incorporates two key components to define module densities: **Type Density** and **Layer Density**, which are combined multiplicatively to determine the final density for each module.

**Type Density.**    Type Density introduces non-uniform sparsity between attention and MLP modules. Based on insights from prior literature (Yuan et al., 2023), MLP modules exhibit higher sensitivity to pruning compared to attention modules. To account for this, we search for the density of attention modules within {Global Density, Global Density $-$ 0.1}, optimizing for performance. The density of MLP modules is then calculated to ensure that the model's global density remains unchanged.

**Layer Density.**    Layer Density accounts for non-uniform sparsity across layers. For this, MPIFA$_{\mathrm{NS}}$ adopts the layerwise density distribution from OWL (Yin et al.), which identifies layer-wise densities based on outlier distribution. By directly utilizing these precomputed layer densities, MPIFA$_{\mathrm{NS}}$ ensures that density is allocated more effectively across layers, balancing pruning across regions of varying importance.

Table 8: **C4 Perplexity (↓) at different parameter density** (proportion of remaining parameters relative to the original model). The best-performing method is highlighted in **bold**.

| Model | Method | Density | | | | | | |
|---|---|---|---|---|---|---|---|---|
| | | 100% | 90% | 80% | 70% | 60% | 50% | 40% |
| LLaMA2-7B | SVD | 7.29 | 18931 | 27154 | 37208 | 56751 | 58451 | 70567 |
| | ASVD | | **7.98** | 12.46 | 201.0 | 9167 | 25441 | 24290 |
| | SVD-LLM | | 13.95 | 19.89 | 33.02 | 61.97 | 129.8 | 262.9 |
| | MPIFA | | 8.15 | **10.20** | **14.68** | **25.43** | **52.01** | **97.71** |
| LLaMA2-13B | SVD | 6.74 | 1994 | 6301 | 37250 | 22783 | 18196 | 84680 |
| | ASVD | | **7.15** | 9.30 | 23.54 | 468.5 | 3537 | 3703 |
| | SVD-LLM | | 10.93 | 14.99 | 24.44 | 46.65 | 110.4 | 267.8 |
| | MPIFA | | 7.27 | **8.79** | **21.00** | **21.33** | **42.03** | **80.47** |
| LLaMA2-70B | SVD | 5.74 | 10.16 | 23.28 | 121.4 | 1659 | 7045 | 12039 |
| | ASVD | | OOM | OOM | OOM | OOM | OOM | OOM |
| | SVD-LLM | | 7.12 | 8.71 | 12.21 | 21.40 | 44.10 | 103.3 |
| | MPIFA | | **6.00** | **6.76** | **8.67** | **13.60** | **29.04** | **63.38** |
| LLaMA3-8B | SVD | 9.47 | 323597 | 461991 | 172968 | 70896 | 143573 | 271176 |
| | ASVD | | **14.43** | 272.1 | 8511 | 18701 | 108117 | 9466 |
| | SVD-LLM | | 38.54 | 98.65 | 223.5 | 460.0 | 784.8 | 1416 |
| | MPIFA | | 14.76 | **22.45** | **44.62** | **123.0** | **257.4** | **429.2** |

**Final Module Density.** The final density for each module in MPIFA$_{\text{NS}}$ is calculated as:

$$\text{Module Density} = \frac{\text{Type Density} \times \text{Layer Density}}{\text{Global Density}}.$$

This formulation ensures that the final density for each module accounts for both type- and layer-specific sparsity requirements, leading to a more effective pruning configuration optimizing performance while maintains the global density same.

In summary, MPIFA$_{\text{NS}}$ combines the benefits of non-uniform sparsity across both types of modules and individual layers, achieving better performance while ensuring the global density of the model remains unchanged.

### B.3. MPIFA$_{\text{NS}}$ Fine-tuning

Fine-tuning is performed using a mixed dataset comprising the training set of WikiText2 and one shard (1/1024) of the training set of C4 (Raffel et al., 2020). WikiText2's training set is more aligned with the evaluation dataset but contains a limited number of tokens, whereas the C4 dataset is significantly larger but less aligned with the test set. To balance these characteristics, the datasets are mixed at a ratio of 2% WikiText2 to 98% C4.

We limit fine-tuning to a single epoch, which requires approximately one day on a single GPU (around 1000 steps). The fine-tuning process updates all pruned parameters, including low-rank matrices and semi-sparse matrices, while keeping other parameters, such as embeddings, fixed.

The learning rate is set to $3 \times 10^{-6}$, with a warmup phase covering the first 5% of total steps, followed by a linear decay to zero. The sequence length is fixed at 1024, with a batch size of 1 and gradient accumulation steps of 128.

## C. Perplexity Evaluation on C4 Dataset

To complement our evaluation on WikiText2, we assess MPIFA on the **C4 dataset** (Raffel et al., 2020), a widely used benchmark for evaluating LLM perplexity on large-scale, real-world text distributions. For consistency with the main text, we use WikiText2 as the calibration dataset for all methods in this evaluation. The results, summarized in Table 8, demonstrate that MPIFA consistently achieves the lowest perplexity across most density levels and model sizes, reducing the

perplexity gap by **47.6%** on LLaMA2-7B, **34.5%** on LLaMA2-13B, **55.3%** on LLaMA2-70B, and **62.6%** on LLaMA3-8B on average across all densities, compared to the best-performing baseline.

## D. Zero-Shot Evaluation on SuperGLUE Benchmark

To provide a more comprehensive evaluation of MPIFA's performance across different compression levels, we conduct zero-shot evaluations at various compression rates on the **SuperGLUE benchmark** (Wang et al., 2019) using the LLaMA2-7B model.

Evaluations are conducted using the `lm-evaluation-harness` framework (Gao et al., 2023). All tasks are evaluated using accuracy (↑), except for the ReCoRD task, which uses F1 score. The detailed accuracy results are presented in Table 9. The best-performing method at each setting is highlighted in **bold**. MPIFA consistently achieves the **highest mean accuracy across all density levels**, outperforming other low-rank methods across a wide range of tasks.

Table 9: **Zero-shot evaluations** on SuperGLUE datasets at different parameter density on LLaMA2-7B. All tasks are evaluated using accuracy (↑), except for the ReCoRD task, which uses F1 score (↑). The best-performing method is highlighted in **bold**.

| Density | Method | BoolQ | CB | COPA | MultiRC | ReCoRD | RTE | WIC | WSC | **Mean** |
|---|---|---|---|---|---|---|---|---|---|---|
| 100% | Dense | 77.7 | 42.9 | 87.0 | 57.0 | 91.6 | 63.2 | 49.7 | 36.5 | 63.2 |
| 90% | SVD | 42.6 | 39.3 | 67.0 | 51.0 | 16.4 | 55.6 | **49.8** | **62.5** | 48.0 |
| | ASVD | 55.9 | 37.5 | 69.0 | 47.1 | 42.5 | 53.4 | **49.8** | 41.3 | 49.6 |
| | SVD-LLM | 49.1 | 41.1 | 79.0 | **57.1** | 87.8 | 52.7 | 48.0 | 48.1 | 57.8 |
| | MPIFA | **74.4** | **64.3** | **86.0** | 56.7 | **91.2** | **58.5** | 49.7 | 36.5 | **64.7** |
| 80% | SVD | 45.9 | **57.1** | 59.0 | 48.9 | 12.5 | 47.3 | 50.0 | 58.7 | 47.4 |
| | ASVD | 41.6 | 33.9 | 58.0 | 46.8 | 24.6 | **55.2** | 50.0 | **60.6** | 46.3 |
| | SVD-LLM | 44.2 | 41.1 | 79.0 | 55.7 | 84.7 | 53.1 | **50.6** | 57.7 | 58.3 |
| | MPIFA | **69.4** | 41.1 | **83.0** | **55.8** | **90.3** | 53.4 | 48.7 | 36.5 | **59.8** |
| 70% | SVD | 40.2 | **46.4** | 62.0 | 43.1 | 12.4 | **53.8** | 48.9 | **63.5** | 46.3 |
| | ASVD | 39.2 | **46.4** | 59.0 | 48.2 | 14.0 | 49.1 | **50.3** | **63.5** | 46.2 |
| | SVD-LLM | 44.4 | 39.3 | **82.0** | 44.6 | 79.8 | **53.8** | 48.9 | 60.6 | 56.7 |
| | MPIFA | **64.8** | 41.1 | 80.0 | **57.2** | **87.5** | **53.8** | 49.1 | 42.3 | **59.5** |
| 60% | SVD | 45.5 | **41.1** | 61.0 | **49.4** | 11.4 | 51.6 | 50.0 | 60.6 | 46.3 |
| | ASVD | 48.9 | 37.5 | 59.0 | 46.6 | 15.2 | 50.2 | 50.0 | 40.4 | 43.5 |
| | SVD-LLM | 38.5 | 39.3 | 71.0 | 44.9 | 66.4 | 53.4 | 50.2 | 59.6 | 52.9 |
| | MPIFA | **56.6** | 35.7 | **79.0** | 44.5 | **82.8** | **57.8** | **52.0** | **63.5** | **59.0** |
| 50% | SVD | 38.6 | **44.6** | 63.0 | 43.0 | 10.1 | **52.7** | 50.2 | **63.5** | 45.7 |
| | ASVD | **42.2** | 39.3 | 63.0 | **45.7** | 14.4 | **52.7** | 50.0 | **63.5** | 46.3 |
| | SVD-LLM | 37.9 | 41.1 | 66.0 | 42.8 | 50.5 | **52.7** | 50.0 | **63.5** | 50.5 |
| | MPIFA | 38.0 | 35.7 | **70.0** | 42.8 | **71.1** | 52.4 | 50.0 | **63.5** | **52.9** |
| 40% | SVD | **49.5** | **42.9** | 61.0 | **48.1** | 11.3 | 52.0 | **50.0** | 49.0 | 45.5 |
| | ASVD | 42.6 | **42.9** | 59.0 | 47.7 | 14.5 | 53.1 | **50.0** | **64.4** | 46.8 |
| | SVD-LLM | 37.8 | 41.1 | 67.0 | 42.8 | 37.0 | 52.7 | **50.0** | 63.5 | 49.0 |
| | MPIFA | 37.8 | 37.5 | **68.0** | 42.8 | **54.7** | **53.8** | **50.0** | 63.5 | **51.0** |

## E. Comparison with Structured Pruning Baseline

To provide a fair comparison with structured pruning methods, we include additional evaluations of **LLM-Pruner** (Ma et al., 2023) as a baseline. All experiments are conducted on the WikiText2 dataset using the LLaMA2-7B model.

**Perplexity Comparison.** Table 10 shows the perplexity across various parameter densities. On average, MPIFA reduces the perplexity gap by **87.2%** compared to LLM-Pruner.

Table 10: **LLM-Pruner vs. MPIFA: Perplexity Comparison** on WikiText2 using LLaMA2-7B at different densities. MPIFA consistently achieves lower perplexity across all densities.

| Method | 100% | 90% | 80% | 70% | 60% | 50% | 40% |
|--------|------|------|------|-------|-------|-------|------|
| LLM-Pruner | 5.47 | 6.58 | 8.81 | 13.70 | 40.49 | 126.0 | 1042 |
| MPIFA | | **5.69** | **6.16** | **7.05** | **8.81** | **12.77** | **21.25** |

**Inference Speedup and Memory Efficiency.** Table 11 and Table 12 compare the inference speedup and memory usage (relative to dense linear layer) for PIFA and LLM-Pruner layers, across different hidden dimensions on an A6000 GPU.

Table 11: **Inference speedup (× over dense)** for PIFA and LLM-Pruner layers.

| Method (density) | d=16384 | d=8192 | d=4096 |
|------------------|---------|--------|--------|
| PIFA (55%) | 1.88× | 1.70× | 1.43× |
| LLM-Pruner (55%) | 1.81× | 1.77× | 1.67× |
| LLM-Pruner (70%) | 1.42× | 1.41× | 1.35× |

Table 12: **Memory usage (× over dense)** for PIFA and LLM-Pruner layers.

| Method (density) | d=16384 | d=8192 | d=4096 |
|------------------|---------|--------|--------|
| PIFA (55%) | 0.56× | 0.58× | 0.64× |
| LLM-Pruner (55%) | 0.56× | 0.58× | 0.65× |
| LLM-Pruner (70%) | 0.70× | 0.72× | 0.75× |

At the same density (55%), PIFA achieves similar speedup and memory efficiency as LLM-Pruner. When comparing MPIFA at 55% density to LLM-Pruner at 70% density, MPIFA consistently achieves lower perplexity, faster inference, and reduced memory usage.

## F. Compression Time and Memory Usage Comparison

We provide a detailed comparison of the compression time and peak GPU memory usage **during compression** across different methods. These metrics reflect the practical resource requirements of each approach.

**Compression Time.** Table 13 reports the time required to perform compression for each method, measured on a single A6000 GPU. On an A100 GPU, the compression time is approximately halved. ASVD requires significantly longer compression time, up to **10–20 hours**, due to the need for sensitivity-based truncation rank searching. The M method involves only reconstruction.

Table 13: **Compression time** for different methods on a single A6000 GPU.

| Model | ASVD | SVD-LLM | PIFA | M |
|-------|------|---------|--------|--------|
| LLaMA2-7B | 10h | 30 min | 15 min | 30 min |
| LLaMA2-13B | 20h | 1h | 30 min | 1h |

**Peak Memory Usage During Compression.** Table 14 reports the peak GPU memory usage during compression. Method M has a relatively low memory footprint, as it processes one layer and one sample at a time.

Table 14: **Peak GPU memory usage (GB)** during compression for different methods.

| Model | ASVD | SVD-LLM | PIFA | M |
|---|---|---|---|---|
| LLaMA2-7B | 15G | 20G | 0.5G | 6G |
| LLaMA2-13B | 30G | 25G | 1G | 10G |

For the **M** method, we report only the reconstruction time, excluding the time required for the initial low-rank pruning step, which could be SVD or SVD-LLM. The low memory footprint of M is primarily attributed to:

- **Online calibration:** Only the current sample is loaded into GPU memory, while the rest of the samples remain in CPU memory until needed.

- **Layer-wise loading:** Only the current pruning layer is loaded to the GPU at any time, with all other layers remaining on CPU.

## G. Enhancing Other Low-Rank Pruning Methods with PIFA and M

To demonstrate the generality of our proposed methods, we evaluate how **PIFA** and **M** can be applied on top of existing low-rank pruning techniques beyond SVD-LLM, including the pruning strategies proposed in the ESPACE paper (Sakr & Khailany).

For a pruning-only comparison, we reproduce the pruning step of ESPACE and compare its performance when combined with PIFA and M. ESPACE introduces six variants: MSE (Eq. 6), MSE-NORM (Eq. 7), GO-MSE (Eq. 8), GO-MSE-NORM (Eq. 8), NL-MSE (Eq. 9), and NL-MSE-NORM (Eq. 9). We exclude the NL-MSE variants as they rely on backpropagation, which is infeasible on memory-constrained GPUs.

Table 15 reports the perplexity (PPL) on WikiText2 at 50% density using the LLaMA2-7B model, showing how PIFA and M further improve various low-rank pruning baselines.

Table 15: **Perplexity (PPL)** on WikiText2 at 50% density using LLaMA2-7B.

| Pruning Method (X) | X | X + PIFA | X + M | X + MPIFA |
|---|---|---|---|---|
| SVD-LLM (W) | 33.27 | 19.64 | 16.55 | **12.77** |
| ESPACE (MSE) | 280.19 | 144.32 | 20.99 | **16.84** |
| ESPACE (MSE-NORM) | 172.30 | 113.82 | 21.73 | **17.42** |
| ESPACE (GO-MSE) | 41.75 | 24.17 | 17.47 | **13.55** |
| ESPACE (GO-MSE-NORM) | 37.45 | 23.19 | 17.47 | **13.63** |

Here, SVD-LLM (W) refers to the standalone pruning output from SVD-LLM without additional reconstruction.

PIFA and M consistently improve the performance of different low-rank pruning methods, including ESPACE, demonstrating that they are general-purpose techniques applicable across various low-rank strategies. These results suggest that PIFA and M are not specific to SVD-LLM but can serve as plug-in modules for other pruning methods to improve accuracy while maintaining efficiency.

