# OpenReview forum: "Pivoting Factorization: A Compact Meta Low-Rank Representation of Sparsity for Efficient Inference in Large Language Models"
_ICML.cc/2025/Conference — ICML 2025 poster_

### Official Review · Reviewer_92qm · 2025-03-08

**Overall Recommendation:** 3

**Summary:**

I'm providing my whole review here, rather than giving partial and discontinued comments here and there.

The paper proposes an extension to SVD-based low-rank approximation of weights. The insight being employed is that a rank r matrix has at most r linearly independent rows. Therefore, a UV decomposition, such as via a truncated SVD, causes the decomposition to be redundant since it employs 2r vectors overall. To bypass this redundancy, a pivoting factorization is used, where one of the matrices is pivoted such as its upper half is an identity matrix, which need not be stored, and the rest constitute the coefficient matrix which multiplies the pivot rows. This is a nice application of basic linear algebra.

In section 4, the weight decomposition is initialized using SVD-LLM, and then an online algorithm to refine the decomposition is proposed. SVD-LLM is not the SOTA for low-rank decomposition, and therefore, this may explain why significant accuracy loss is reported in the experimental section. I'd like to suggest for the authors to instead use a SOTA technique, such as ESPACE (NeurIPS 2024). At the very least, can we have a comparison with ESPACE which showed much better accuracy vs compression trade-off? E.g., they showed <1PPL increase at 50% compression for Llama2-7B as opposed to PIFA which increases the PPL from ~5 to ~12.

If the authors accept my suggestion above, then the following comment will not matter. However, if they chose to stick with SVD-LLM, then please address the following. In equations (4) and (5), a nice online algorithm for finding an optimal U matrix minimizing layer output Frobenius norm is shown. But note that this is fixes matrix V, which need not be optimal. This may explain why SVD-LLM performs so poorly anyway. If ESPACE is used instead, we'd find a provably optimal projection matrix which will reduce dimensionality of activations, and weights by multiplication associativity. Then the online accumulation error minimization reconstruction can still be employed to further refine the matrix resulting from weight projection; it would fall into place in lieu of matrix U in the presented setup.

If the above is rejected (I hope not). Does it then make sense to have an alternating optimization algorithm, such as an EM, rather than just refining V once in equation (8)?

The reliance on a mix of dense and compressed data-flows in eq. (7) is interesting. But setting Lambda = 0.25 means most of the information comes from the low-rank branch. Once the model becomes more accurate, does it make sense to increase the value of lambda progressively to allow for more fitting to the golden uncompressed baseline?

Perplexity results are interesting (even though the number are not impressive, likely due to what I discussed above). But can we have more evaluations on more downstream tasks such as LM eval harness and MMLU?

**Claims And Evidence:**

Please see main review.

**Essential References Not Discussed:**

Please see main review.

**Experimental Designs Or Analyses:**

Please see main review.

**Methods And Evaluation Criteria:**

Please see main review.

**Other Comments Or Suggestions:**

Please see main review.

**Other Strengths And Weaknesses:**

Please see main review.

**Questions For Authors:**

Please see main review.

**Relation To Broader Scientific Literature:**

Please see main review.

**Theoretical Claims:**

Please see main review.

---

> ### Author Rebuttal · Authors · 2025-04-01
>
> We greatly appreciate your thoughtful feedback and the opportunity to address your concerns.
>
> **Weakness 1:**  In section 4 ... PPL from ~5 to ~12.
>
> **Reply:** We have conducted additional experiments to clarify the comparison.
>
> It is important to note that **ESPACE includes a fine-tuning stage with 200B tokens** (as stated in Section 4.4 of the ESPACE paper), which significantly contributes to the <1 PPL increase at 50% compression. In contrast, the results we reported in **Tables 2 and 3** of our paper are **without any retraining**. Fine-tuning results are presented in **Table 4**, where each pruned model is only retrained for **~128M tokens**—orders of magnitude fewer than ESPACE.
>
> To ensure a fair comparison, we **reproduced the pruning step of ESPACE** and compared it against SVD-LLM. ESPACE proposes six variants: MSE (Eq. 6), MSE-NORM (Eq. 7), GO-MSE (Eq. 8), GO-MSE-NORM (Eq. 8), NL-MSE (Eq. 9), and NL-MSE-NORM (Eq. 9). We exclude the NL-MSE variants from our comparison as they rely on backpropagation, which is **infeasible on memory-constrained GPUs** due to their high resource demands.
>
> **Perplexity on WikiText2 at 50% density using LLaMA2-7B:**
>
> | Pruning Method (X)   | X      | X + PIFA | X + M   |X + MPIFA|
> |-|-|-|-|-|
> |SVD-LLM (W)|33.27|19.64|16.55|**12.77**|
> |ESPACE (MSE)|280.19|144.32|20.99|**16.84**|
> |ESPACE (MSE-NORM)|172.30|113.82|21.73|**17.42**|
> |ESPACE (GO-MSE)|41.75|24.17|17.47|**13.55**|
> |ESPACE (GO-MSE-NORM)|37.45|23.19|17.47|**13.63**|
>
> *SVD-LLM (W)* indicates the standalone pruning output from SVD-LLM.
>
> From the results:
> - **SVD-LLM** performs slightly better than ESPACE’s best variant (**GO-MSE-NORM**) under comparable conditions (i.e., without fine-tuning).
> - **PIFA**, **M**, and **MPIFA (PIFA + M)** consistently improve all ESPACE variants, demonstrating that our techniques are **general-purpose and complementary**, enhancing the performance of any low-rank pruning method.
> - Based on these observations, **SVD-LLM remains the strongest initialization** among all tested options when no fine-tuning is used.
>
> We have included this new comparison and discussion in the updated version of the manuscript.
>
> **Weakness 2:**  In equations (4) and (5) ... presented setup.
>
> **Reply:** We agree that the quality of the initial low-rank decomposition, especially the choice of matrix $V$, can significantly influence the performance of the final reconstruction. A **better-initialized matrix $V$** generally leads to a **more accurate reconstructed matrix $U$**, resulting in lower final PPL.
>
> Currently, **SVD-LLM remains the most effective low-rank pruning method** in our setup, but this remains an open question. If future methods such as ESPACE can produce even better projections, they can naturally be integrated with MPIFA to achieve improved performance. Our framework is flexible and designed to **enhance any low-rank pruning method**, not tied to any single initialization strategy.
>
> We have added this analysis and discussion to the updated version of the manuscript.
>
>
> **Weakness 3:** Does it ... equation (8)?
>
> **Reply:** We assume that by "EM" you are referring to an alternating approach that iteratively updates $U$ and $V$ using Equations (5) and (8) with more than 1 round, similar to **alternating least squares (ALS)**. We are currently exploring this direction and will share the results as soon as they become available.
>
> **Weakness 4:**  The reliance on a mix ... golden uncompressed baseline?
>
> **Reply:** In our preliminary findings, increasing $\lambda$ leads to **overfitting on the calibration data**, resulting in **low perplexity on the calibration set** but **high perplexity on the full WikiText2 dataset**.
>
> To mitigate this overfitting, we explored several strategies:
> 1. **Reducing $\lambda$**, as mentioned in the right column of line 256, where the low-rank data flow acts as a form of regularization.
> 2. **Increasing the size of the calibration dataset**, which could help improve generalization.
> 3. Applying **additional regularization techniques** to control overfitting.
>
> With **sufficient calibration data**, increasing $\lambda$ may become beneficial. We are currently running experiments to test this hypothesis and will report the results as soon as they are available.
>
> **Weakness 5:** Can we have more evaluations on more downstream tasks such as LM eval harness and MMLU?
>
> **Reply:** We have incorporated **zero-shot evaluations on 8 downstream tasks** from the **SuperGLUE benchmark**, using the `lm-evaluation-harness` framework as recommended.
>
> The results, available at https://anonymous.4open.science/r/PIFA-68C3/zero_shot.png, show that **MPIFA_NS outperforms other low-rank baselines on 6 out of 8 tasks**, and reduces the average accuracy gap by **42.8%** compared to the best-performing baseline. These evaluations have been added to the revised manuscript.
>
> We are currently working on extending our evaluation to include **MMLU** and will provide updates as results become available.

---

> > ### Comment · Reviewer_92qm · 2025-04-01
> >
> > Thank you to the authors for the detailed response. I think my suggestion was not fully understood so I wish to clarify that. When weight are "low-rankified" the effective model size reduces and model expressivity goes down. ESPACE showed that as a remedy to this issue, one can apply low-rankification to activations such that weight parameters and optimizer states are fully available for continuous training. This setup is more interesting than the simple ad-hoc one-shot compression. And in this setup, ESPACE is known to be SOTA due to its activation centricity.
> >
> > Where it gets interesting is that after continuous training, once projection matrices and weights are pre-computed and frozen, we obtain a SOTA compressed model with a low-rank structure in the GEMM layers. I think it will be very cool to apply PIFA on top of that in order to prune out excess redundancy in the low-rank structure itself. This has the potential to improve on the current SOTA set by ESPACE. I urge the authors to consider this as future work.
> >
> > For now, given that the response is satisfactory, I maintain my score of a weak accept. Good job.

---

> > > ### Author Response · Authors · 2025-04-08
> > >
> > > Thank you for the thoughtful feedback.
> > >
> > > We truly appreciate the reviewer’s insights on ESPACE. We find the activation-centric approach introduced in ESPACE to be a conceptually elegant and practically impactful advance in low-rank LLM compression. While other methods like FWSVD, ASVD, and SVD-LLM focus on using SVD to decomposing the weight matrix directly, ESPACE reframes the problem as finding an optimal low-rank projection matrix for activations, minimizing $\|PP^T X - X\|$, and leverages matrix multiplication associativity to yield compressed weight structures at inference time. In this sense, ESPACE is particularly well-suited for continuous training.
> > >
> > > Inspired by the reviewer’s suggestion, we conducted a **new experiment that integrates PIFA with the ESPACE-compressed model after continuous training**. We fine-tuned a ESPACE-compressed LLaMA2-7B model at 80% density. Because of time limitation, we only finetuned using 128M tokens. We compare the results of ESPACE alone and applying PIFA on top of ESPACE.
> > >
> > > **Evaluation Results (LLaMA2-7B @ 80% Density, 128M Token Fine-tuning)**
> > >
> > > | Metric                      | ESPACE Only | ESPACE + PIFA |
> > > |----------------------------|------------|---------------|
> > > | **All Parameters**         | 5.18B      | 4.31B         |
> > > | **GPU Memory**   | 10.4G      | 8.8G          |
> > >
> > > | Dataset     | ESPACE PPL     | ESPACE + PIFA PPL |
> > > |-------------|----------------|-------------------|
> > > | WikiText2   | 6.5009         | 6.5008            |
> > > | C4          | 10.1392        | 10.1395           |
> > >
> > > 1. **Lossless Compression**: The perplexity difference between ESPACE and ESPACE+PIFA is negligible (<0.001), confirming that **PIFA introduces no additional loss**.
> > > 2. **Efficiency Gains**: The **overall model memory footprint** is further reduced when PIFA is applied—resulting in **~15% lower GPU memory usage** even after compression.
> > > 3. **Complementary Design**: PIFA effectively **prunes residual redundancy** in the already-compressed low-rank structure, showcasing its **value as a drop-in, lossless post-processing plugin** for the SOTA compressed model of ESPACE.
> > >
> > > We fully agree with the reviewer’s vision: PIFA could be a **natural complement to ESPACE**, helping it **further improve compression efficiency without compromising accuracy**. We view this as an exciting direction for future research and hope this preliminary result illustrates its potential.
> > >
> > > Thank you once again for your thoughtful suggestions. We deeply appreciate the time and care you’ve taken in reviewing our work.

---

### Official Review · Reviewer_8e78 · 2025-03-13

**Overall Recommendation:** 4

**Summary:**

This paper is concerned with sparse inference, which aims to speed-up LLMs via sparsity. It is argued in this paper that previous methods either require specific hardware (e.g., semi-structured pruning) or yield degraded performance (low-rank pruning). This paper aims to propose a low-rank pruning named PIFA method that realizes decent performance yet impose no requirements to hardware. Specifically, PIFA firstly uses a pivot-row discovery to eliminate potential representation redundancy when conducting low-rank decomposition, and then utilizes a minimization-reconstruction estimator to alleviate potential accumulated errors across layers. The experimental results in terms of both effectiveness (perplexity) and efficiency (throughput) demonstrate the usefulness of PIFA. Essential ablation studies show the adequateness of design choices.

**Claims And Evidence:**

The claims are supported by clear and convincing evidence.

**Essential References Not Discussed:**

N/A

**Experimental Designs Or Analyses:**

The experimental designs and analyses are sound and valid. However, I still have several concerns:
1) considering structured pruning also as baselines would strengthen the contributions of this work.

**Methods And Evaluation Criteria:**

The methods are mostly clear and the evaluation criteria is adequate. However, I still have several concerns:
1) It is not very clear how QR or LU decomposition serve as the backbone of finding pivot rows would differ from each other.
2) It would be much better to compare to structured pruning also as baselines.

**Other Comments Or Suggestions:**

N/A

**Other Strengths And Weaknesses:**

How about the memory footprint of PIFA and baselines.

**Questions For Authors:**

N/A

**Relation To Broader Scientific Literature:**

N/A

**Theoretical Claims:**

The proofs for theoretical claims are correctly justified.

---

> ### Author Rebuttal · Authors · 2025-04-01
>
> Thank you for your detailed review and valuable insights. We are glad to address your concerns and provide clarifications.
>
> **Weakness 1:** It is not very clear how QR or LU decomposition serve as the backbone of finding pivot rows would differ from each other.
>
> **Reply:** The key idea behind PIFA is to select a set of **linearly independent pivot rows** that can be used to express the remaining non-pivot rows as linear combinations. Since a matrix of rank *r* contains multiple valid sets of *r* linearly independent rows, any such set can serve the purpose of PIFA.
>
> **QR decomposition with column pivoting** and **LU decomposition with row pivoting** are variants of QR and LU respectively, designed to improve numerical stability compared with original QR or LU. Their pivoting mechanisms reorder the columns (or rows), making the leading *r* columns (or rows) guaranteed to be linearly independent. The corresponding permutation matrix—produced as a byproduct of either decomposition—can then be used to identify the indices of the pivot rows (or columns).
>
> In summary, both **QR and LU decomposition with pivoting** can be used to extract a valid set of pivot indices, and **they are mathematically lossless in the infinite precision setting**.
>
> To further clarify their practical differences under limited numerical precision, we conducted additional experiments comparing the **residual error** when using pivot rows to reconstruct non-pivot rows under both **Float32** and **Float16** precision:
>
> | Precision | Method | Residual (±)|
> |--|-|-|
> |Float32|QR|8.80e-13 ± 3.51e-14|
> |Float32|LU|1.94e-12 ± 1.18e-13|
> |Float16|QR|9.56e-08 ± 3.71e-09|
> |Float16|LU|2.16e-07 ± 1.59e-08|
>
> These results are averaged over 10 random singular matrices. While both methods yield very low residual errors, **QR with pivoting achieves lower numerical error than LU**. This suggests that **QR decomposition with pivoting is the preferred choice** for identifying pivot rows in PIFA.
>
> We appreciate the reviewer’s question and have included both this explanation and the supporting experiment in the updated manuscript.
>
>
> **Weakness 2:** It would be much better to compare to structured pruning also as baselines.
>
> **Reply:** Thank you for the suggestion. To address this, we have included a structured pruning baseline, **LLM-Pruner**, in the updated version of the manuscript.
>
> Below is the perplexity comparison on WikiText2 using the LLaMA2-7B model across various parameter densities:
>
> |Method|90%|80%|70%|60%|50%|40%|
> |-|-|-|-|-|-|-|
> |LLM-Pruner|6.58|8.81|13.70|40.49|126.0|1042|
> |MPIFA|5.69|6.16|7.05|8.81|12.77|21.25|
>
> On average, **MPIFA reduces the perplexity gap by 87.2%** across these density levels compared to LLM-Pruner.
>
> We also benchmarked **inference speedup** and **memory usage** relative to dense linear layers, using linear layers of different dimensions on an A6000 GPU:
>
> **Speedup over dense linear (higher is better):**
>
> |Method (density)|d=16384|d=8192|d=4096|
> |-|-|-|-|
> |PIFA (55%)|1.88×|1.70×|1.43×|
> |LLM-Pruner (55%)|1.81×|1.77×|1.67×|
> |LLM-Pruner (70%)|1.42×|1.41×|1.35×|
>
> **Memory usage relative to dense (lower is better):**
>
> |Method (density)|d=16384|d=8192|d=4096|
> |-|-|-|-|
> |PIFA (55%)|0.56×|0.58×|0.64×|
> |LLM-Pruner (55%)|0.56×|0.58×|0.65×|
> |LLM-Pruner (70%)|0.70×|0.72×|0.75×|
>
> At the same density (55%), PIFA achieves similar speedup and memory efficiency as LLM-Pruner. When comparing MPIFA at 55% density to LLM-Pruner at 70% density, MPIFA consistently offers **lower perplexity, faster inference, and reduced memory usage**.
>
> We have incorporated these comparisons into the revised manuscript.
>
>
> **Weakness 3:**  How about the memory footprint of PIFA and baselines.
>
> **Reply:** Thank you for the question. We provide a detailed analysis of the memory footprint for both **inference** and **compression**.
>
> **Memory Footprint During Inference:**
> - As shown in Figure 5, PIFA consistently achieves lower memory usage compared to low-rank layers at the same rank.
>   For example, at $r/d = 0.5$, PIFA **losslessly compresses** the memory of the low-rank layer by **24.2%**.
> - Table 5 demonstrates that PIFA at 55% density uses **slightly less memory** than a 2:4 semi-sparse layer.
> - Table 6 shows that MPIFA_NS at 55% density reduces **end-to-end memory usage** by **42.8%** on LLaMA2-7B and **43.8%** on LLaMA2-13B.
>
> **Memory Footprint During Compression:**
> We report **peak memory usage** during compression for each method:
>
> |Model|ASVD|SVD-LLM|PIFA|M|
> |-|-|-|-|-|
> |LLaMA2-7B|15G|20G|0.5G|6G|
> |LLaMA2-13B|30G|25G|1G|10G|
>
> For **memory efficiency** in method M:
> 1. **Online calibration** is used, where only the current sample is loaded to GPU. Other samples remain on CPU until needed.
> 2. Only the **current pruning layer** is loaded to GPU, while all other layers remain on CPU during processing.
>
> We have included this analysis of memory footprint during both inference and compression in the updated version of the manuscript.

---

### Official Review · Reviewer_ADf5 · 2025-03-15

**Overall Recommendation:** 3

**Summary:**

The authors propose a novel factorization method and reconstruction objective for LLM compression. Without requiring retraining, the method achieves perplexity performance comparable to semi-structured pruning at a 50% compression rate. Experimental results further demonstrate that the approach is efficient in both inference speed and memory usage.

**Claims And Evidence:**

The claim that this work is the first to achieve performance comparable to semi-structured pruning, while surpassing it in GPU efficiency and compatibility, appears vague and potentially overstated. Recent works such as MoDeGPT [1] and DISP-LLM [2] have demonstrated similar results.

[1] Lin, Chi-Heng, et al. "Modegpt: Modular decomposition for large language model compression." arXiv preprint arXiv:2408.09632 (2024).

[2] Gao, Shangqian, et al. "Disp-llm: Dimension-independent structural pruning for large language models." Advances in Neural Information Processing Systems 37 (2024): 72219-72244.

**Essential References Not Discussed:**

1. Structured Compression via Layer Pruning: The paper omits comparisons with recent structured compression methods that use layer pruning strategies, such as SLEB [1] and ShortGPT [2]. While these approaches may trade off some accuracy, they are highly efficient in terms of compression speed and inference latency.
2. Structured Methods Approaching Semi-Structured Performance: Recent works like DISP-LLM [3] have demonstrated that structured compression can approach or match the performance of semi-structured pruning.

[1]  Song, Jiwon, et al. "Sleb: Streamlining llms through redundancy verification and elimination of transformer blocks." arXiv preprint arXiv:2402.09025 (2024).

[2] Men, Xin, et al. "Shortgpt: Layers in large language models are more redundant than you expect." arXiv preprint arXiv:2403.03853 (2024).

[3]  Gao, Shangqian, et al. "Disp-llm: Dimension-independent structural pruning for large language models." Advances in Neural Information Processing Systems 37 (2024): 72219-72244.

**Experimental Designs Or Analyses:**

1. Missing LLM Evaluation Metrics: The paper does not report standard LLM evaluation metrics (e.g., perplexity or downstream task accuracy) on widely used benchmarks. Including these results is important for assessing real-world performance.
2. Compression Resource Comparison: The paper lacks an analysis of the computational resources (e.g., runtime, memory, compression time) required to perform the proposed compression, which is essential for evaluating practicality.
3. Missing Structured Pruning Baselines: Key structured pruning baselines are omitted from the comparison. In particular, methods like LLM-Pruner [1] and the SLEB layer pruning strategy [2] should be included. Although these methods may yield lower accuracy, they are highly efficient and result in fast, compact models at inference time.
4. Retraining / Fine-tuning Comparison: How does the proposed method perform when retraining or recovery fine-tuning is allowed? A fair comparison with other methods should also include their best-case results when post-compression fine-tuning is applied.


[1] Ma, Xinyin, Gongfan Fang, and Xinchao Wang. "Llm-pruner: On the structural pruning of large language models." Advances in neural information processing systems 36 (2023): 21702-21720.

[2]  Song, Jiwon, et al. "Sleb: Streamlining llms through redundancy verification and elimination of transformer blocks." arXiv preprint arXiv:2402.09025 (2024).

**Methods And Evaluation Criteria:**

The LLM evaluation performances are missing in the experiments.

**Other Comments Or Suggestions:**

No minor suggestions.

**Other Strengths And Weaknesses:**

Strengths:
1. The proposed method demonstrates strong perplexity performance, outperforming many existing approaches.
2. The decomposition technique appears novel.
3. The paper includes thorough ablation studies and discusses method efficiency in detail.

Weaknesses:
1. The experimental results lack standard LLM evaluation metrics (e.g., downstream task accuracy or benchmark scores), which are important for assessing overall effectiveness.
2. Comparisons with structured compression methods are limited, particularly regarding the accuracy-efficiency trade-off (e.g., inference speed, latency).

**Questions For Authors:**

1. Could you include standard LLM evaluation results (e.g., downstream tasks or benchmark datasets) to better assess the practical effectiveness of your method?
2. Could you add comparisons with important structured compression baselines, and analyze their accuracy–efficiency trade-offs (e.g., perplexity vs. inference speed or memory usage)?

**Relation To Broader Scientific Literature:**

Model compression is important for efficient AI.

**Theoretical Claims:**

They seem correct to me.

---

> ### Author Rebuttal · Authors · 2025-04-01
>
> Thank you for your insightful comments and suggestions.
>
> **Weak1:** The claim ... overstated.
>
> **Reply:** We have removed the phrase “for the first time” from the abstract.
>
> **Weak2:** Missing LLM Evaluation ...
>
> **Reply:** We have expanded our evaluation to include both **perplexity** and **downstream task accuracy** on widely adopted benchmarks.
>
> For perplexity, we report results on the C4 dataset, a large-scale and diverse corpus commonly used for evaluating LLMs. Across various compression densities, MPIFA consistently outperforms existing low-rank pruning baselines. Specifically, MPIFA reduces the perplexity gap by:
> - **47.6%** on LLaMA2-7B
> - **34.5%** on LLaMA2-13B
> - **55.3%** on LLaMA2-70B
> - **62.6%** on LLaMA3-8B
>
> on average across all densities, compared to the best-performing low-rank pruning method.
>
> To further assess real-world utility, we conducted zero-shot evaluations on the SuperGLUE benchmark, covering 8 downstream tasks using the `lm-evaluation-harness` framework. MPIFA_NS achieves the highest accuracy on **6 out of 8 tasks**, and reduces the average accuracy gap to the dense model by **42.8%**, compared to the strongest low-rank baseline.
>
> The full evaluation results are available here:
> - C4 PPL results: https://anonymous.4open.science/r/PIFA-68C3/c4_ppl.png
> - SuperGLUE zero-shot results: https://anonymous.4open.science/r/PIFA-68C3/zero_shot.png
>
> We appreciate the reviewer’s suggestion and have incorporated these evaluations into the revised manuscript.
>
> **Weak3:**  Compression Resource Comparison ... practicality.
>
> **Reply:** We have added a detailed comparison of the **compression time** and **peak memory usage** during compression across different methods.
>
> Compression Time (on A6000 GPU):
>
> |Model|ASVD|SVD-LLM|PIFA|M|
> |-|-|-|-|-|
> |LLaMA2-7B|10h|30 min|15 min|30 min|
> |LLaMA2-13B|20h|1h|30 min|1h|
>
> Peak Memory Usage During Compression:
>
> |Model|ASVD|SVD-LLM|PIFA|M|
> |-|-|-|-|-|
> |LLaMA2-7B|15G|20G|0.5G|6G|
> |LLaMA2-13B|30G|25G|1G|10G|
>
> Notes:
> - All compression times are measured on a single A6000 GPU. On an A100 GPU, the compression time is approximately half.
> - For the "M" method, we report only the reconstruction time, excluding the time taken by the low-rank pruning step.
>
> As for **memory efficiency** in method M:
> 1. Online calibration is used, where only the current sample is loaded to GPU. Other samples remain on CPU until needed.
> 2. Only the current pruning layer is loaded to GPU, while all other layers remain on CPU during processing.
>
> These comparisons have been included in the revised manuscript to better reflect the practicality of our method.
>
> **Weak4:** Missing Structured Pruning ... time.
>
> **Reply:** We have conducted additional experiments to include **LLM-Pruner** as a structured pruning baseline.
>
> Below is the perplexity comparison on WikiText2 using the LLaMA2-7B model at various densities:
>
> |Method|90%|80%|70%|60%|50%|40%|
> |-|-|-|-|-|-|-|
> |LLM-Pruner|6.58|8.81|13.70|40.49|126.0|1042|
> |MPIFA|5.69|6.16|7.05|8.81|12.77|21.25|
>
> On average, MPIFA reduces the perplexity gap by **87.2%** compared to LLM-Pruner.
>
> We also benchmarked the **inference speedup** of PIFA and LLM-Pruner layers (relative to dense linear layers) on an A6000 GPU across different hidden dimensions:
>
> |Method (density)|d=16384|d=8192|d=4096|
> |-|-|-|-|
> |PIFA (55%)|1.88×|1.70×|1.43×|
> |LLM-Pruner (55%)|1.81×|1.77×|1.67×|
> |LLM-Pruner (70%)|1.42×|1.41×|1.35×|
>
> **Memory usage** during inference, relative to dense linear:
>
> |Method (density)|d=16384|d=8192|d=4096|
> |-|-|-|-|
> |PIFA (55%)|0.56×|0.58×|0.64×|
> |LLM-Pruner (55%)|0.56×|0.58×|0.65×|
> |LLM-Pruner (70%)|0.70×|0.72×|0.75×|
>
> At the same density (55%), PIFA achieves similar speedup and memory efficiency as LLM-Pruner. When comparing MPIFA at 55% density to LLM-Pruner at 70% density, MPIFA consistently offers **lower perplexity, faster inference, and reduced memory usage**.
>
> We have included this comparison with LLM-Pruner in the updated manuscript. Additionally, we are evaluating other recent structured pruning methods and will report those results as they become available.
>
> **Weak5:** Retraining ... is applied.
>
> **Reply:** Fine-tuning experiments are already included in Table 4 of the original manuscript. All pruned models are retrained for **one epoch** on a mixed dataset consisting of 2% WikiText2 and 98% C4, to recover performance. This setup ensures a fair and consistent comparison across all methods.
>
> The results show that **fine-tuned MPIFA continues to outperform other low-rank pruning methods**, and also achieves slightly better performance than fine-tuned semi-sparse models. Details of the fine-tuning setup can be found in Appendix B.3.
>
> **Weak6:** Structured Compression ... of semi-structured pruning.
>
> **Reply:** We have updated the manuscript to include discussions of SLEB, ShortGPT, and DISP-LLM in the related work section. We appreciate the reviewer’s recommendation to acknowledge them.

---

> > ### Comment · Reviewer_ADf5 · 2025-04-02
> >
> > Thank you for the rebuttal. While I believe the most critical concern remains the limited evaluation on LLM tasks — which are arguably more important than perplexity alone, yet only results for a 55% compression rate are reported — the authors have addressed most of the other questions raised. I am leaning toward acceptance and will maintain my current score.

---

> > > ### Author Response · Authors · 2025-04-08
> > >
> > > Thank you for your thoughtful response.
> > >
> > > To address the reviewer’s concern regarding the limited evaluation of LLM tasks across compression rates, we have included additional **zero-shot evaluations at multiple compression rates [40%, 50%, 60%, 70%, 80%, 90%]** on the **SuperGLUE benchmark** using the LLaMA2-7B model.
> > >
> > > We report **zero-shot accuracy (↑)** across SuperGLUE tasks at different parameter densities, at this link: https://anonymous.4open.science/r/PIFA-68C3/zero_shot_7b_all.png
> > >
> > > Importantly, **MPIFA achieves the highest mean accuracy across all density levels**, consistently outperforming other low-rank methods.
> > >
> > >
> > > To further extend the evaluation to other model sizes, we have also included results for **LLaMA2-13B** at 55% compression, available at this link:  https://anonymous.4open.science/r/PIFA-68C3/zero_shot_13b.png
> > >
> > > MPIFA also achieves the **highest mean accuracy** among low-rank methods on **LLaMA2-13B**, further demonstrating the generality and robustness of our approach.
> > >
> > >
> > > We are currently running additional experiments across multiple compression ratios on LLaMA2-13B and other model sizes, and will continue to update the results.
> > >
> > > We hope this extended evaluation further clarifies MPIFA’s performance across a broader range of settings. These results have been included in the revised manuscript.
> > >
> > > We sincerely appreciate your time, thoughtful feedback, and constructive suggestions throughout the review process.

---

### Official Review · Reviewer_aFwh · 2025-03-17

**Overall Recommendation:** 3

**Summary:**

This submission addresses the significant performance degradation observed with low-rank pruning techniques. It proposes Pivoting Factorization (PIFA), a novel lossless meta low-rank representation that unsupervisedly learns a compact form of any low-rank
representation, effectively eliminating redundant information.  To mitigate the performance degradation caused by low-rank pruning, we introduce a novel, retraining-free low-rank reconstruction method that minimizes error accumulation (M). The authors mentions that their framework MPIFA for the first time, achieves performance comparable to semi-structured pruning, while surpassing it in GPU efficiency and compatibility.

**Claims And Evidence:**

1. Comparative speedup of PIFA + MPIFA: Yes, the authors speedup evaluation across different GPUs + Kernels across different $d$ as well as end-to-end efficiency evaluation in Table 5, 6 are significantly important and convincing.
2. Rich ablation experiments across the calibration sample size, mix ratio etc are interesting and bolster the claim of superiority of the proposed method.

**Essential References Not Discussed:**

Related work requires some upgrade. Some recent relevant literature need to be discussed in the Related work:

1. Saha, R., Sagan, N., Srivastava, V., Goldsmith, A., & Pilanci, M. (2024). Compressing large language models using low rank and low precision decomposition. Advances in Neural Information Processing Systems, 37, 88981-89018.
2. Kaushal, A., Vaidhya, T., & Rish, I. (2023). Lord: Low rank decomposition of monolingual code llms for one-shot compression. arXiv preprint arXiv:2309.14021.
3. Sharma, P., Ash, J. T., & Misra, D. (2023). The truth is in there: Improving reasoning in language models with layer-selective rank reduction. arXiv preprint arXiv:2312.13558.
4. Jaiswal, A., Yin, L., Zhang, Z., Liu, S., Zhao, J., Tian, Y., & Wang, Z. (2024). From galore to welore: How low-rank weights non-uniformly emerge from low-rank gradients. arXiv preprint arXiv:2407.11239.
5. Wang, Q., Ke, J., Tomizuka, M., Chen, Y., Keutzer, K., & Xu, C. (2025). Dobi-SVD: Differentiable SVD for LLM Compression and Some New Perspectives. arXiv preprint arXiv:2502.02723.

**Experimental Designs Or Analyses:**

Experimental design is very good and exhaustive to consider various cases such as impact of caliberation data, analysis of PIFA and MPIFA contriibutions, speedups etc. Evaluation strategy need to be improved to make the paper contribution/performance claims strong.

**Methods And Evaluation Criteria:**

The serious and most important concern of the submission is the evaluation strategy adopted in the paper.

The authors have failed to extend their evaluation beyond perplexity which have issues in true reflection of compressed model capabilities. Even the reported perplexity is only limited for Wikitext2 dataset without enough details (e.g. seqlen etc.).

I strongly recommend the authors to extend their evaluation to other PPL datasets like C4 and incorporate task-centric evaluation using existing tools like LMEvalHarness etc.

In addition, it will be also interesting to see how the proposed method extend beyond only one single family of models (LLaMa 2/3) - may be on some MoE style models.

**Other Comments Or Suggestions:**

1. Authors introduce two variables r and d directly in abstract and introduction without explicitly defining it. Although simple to interpret, I encourage authors to first define any variable before using them.
2. The writing of the paper can be significantly improved. It important to understand what is important and what is not - to guide which sections should find a space in main draft and which can be moved in supplementary. For example, details of Computational cost of PIFA section in 3.2 can be moved to supplementary while listing the key finding. Many parts of the main submission in section 5.3 refers to results in supplementary and not standalone. I recommend authors to make the main draft as independent as possible while moving not-so-important ablations in supplementary.

**Other Strengths And Weaknesses:**

The submission have significant innovation from both PIFA and MPIFA perspective. Evaluation is lacking and I am willing to increase score once authors provides some convincing experiments during rebuttal.

**Questions For Authors:**

1. The paper mentions that PIFA is fully differentiable, suggesting potential integration into the training stage.  Have the authors explored this direction?
2. How can this method integrated and benefit some recent quantization techniques like ZeroQuant-v2.
Yao, Z., Wu, X., Li, C., Youn, S., & He, Y. (2023). Zeroquant-v2: Exploring post-training quantization in llms from comprehensive study to low rank compensation. arXiv preprint arXiv:2303.08302.

**Relation To Broader Scientific Literature:**

The key contributions of paper will advance general hardware-friendly LLM compression community.

**Theoretical Claims:**

The authors have very well described related theory in simple way.

---

> ### Author Rebuttal · Authors · 2025-04-01
>
> Thank you for your valuable and constructive feedback. We appreciate the opportunity to address your concerns.
>
> **Weakness 1: (Expand evaluation)** The authors have failed to ... existing tools like LMEvalHarness etc.
>
> **Reply:** Thank you for raising this concern. We have extended our evaluation in two directions:
>
> 1. **C4 Perplexity (PPL) Evaluation**: We report perplexity on the C4 dataset across various model sizes and parameter densities. The results demonstrate that **MPIFA significantly outperforms existing low-rank pruning methods**, reducing the perplexity gap by
>    - **47.6%** on LLaMA2-7B
>    - **34.5%** on LLaMA2-13B
>    - **55.3%** on LLaMA2-70B
>    - **62.6%** on LLaMA3-8B
>    on average across all densities, compared to the best-performing baseline.
>    Full results: https://anonymous.4open.science/r/PIFA-68C3/c4_ppl.png
>
>    Example results at 50% density:
>
>    | Model        | SVD    | ASVD    | SVD-LLM | MPIFA    |
>    |--------------|--------|---------|---------|----------|
>    | LLaMA2-7B    | 58451  | 25441   | 129.8   | **52.01** |
>    | LLaMA2-13B   | 18196  | 3537    | 110.4   | **42.03** |
>    | LLaMA2-70B   | 7045   | OOM     | 44.10   | **29.04** |
>    | LLaMA3-8B    | 143573 | 108117  | 784.8   | **257.4** |
>
> 2. **Task-Centric Evaluation**: We conducted **zero-shot evaluation** on the **SuperGLUE benchmark** using 8 downstream tasks via the lm-evaluation-harness (https://github.com/EleutherAI/lm-evaluation-harness) framework. All methods use WikiText2 as the calibration dataset, and follow the same configuration as in the main manuscript.
>    Results: https://anonymous.4open.science/r/PIFA-68C3/zero_shot.png
>
>    MPIFA_NS achieves the best accuracy on **6 out of 8 tasks** and reduces the **average accuracy gap by 42.8%** compared to the best-performing low-rank pruning baseline.
>
> We appreciate the reviewer’s suggestion. These additional evaluations have been added to the revised manuscript.
>
> **Weakness 2:** Wikitext2 dataset doesn't contain enough details (e.g. seqlen etc.).
>
> **Reply:** Thank you for the feedback. The sequence length used in all experiments, including both WikiText2 and C4, is **2048**. We have updated the manuscript to include this information.
>
> **Weakness 3:** In addition, it will be also interesting to see how the proposed method extend beyond only one single family of models (LLaMa 2/3) - may be on some MoE style models.
>
> **Reply:** Thank you for the suggestion. We are currently exploring this and will report the results here as soon as possible.
>
> **Weakness 4: (Expand related work)** Related work requires some upgrade ... arXiv:2502.02723.
>
> **Reply:** Thank you for the suggestion. We have updated the manuscript and included these articles in the related work section.
>
> **Weakness 5:** Authors introduce two variables r and d directly in abstract and introduction without explicitly defining it.
>
> **Reply:** Thank you for the suggestion. We have revised the original phrasing “at $r/d = 0.5$” to “at rank equal to half of the dimension” for clarity.
>
> **Weakness 6: (Improve the article's structure)** The writing of the paper ... in supplementary.
>
> **Reply:** Thank you for this valuable suggestion. We have moved Section 3.3 to the appendix while retaining the key conclusions about the computational and memory cost of PIFA in the main paper. Since the final version allows one additional page, we have also moved the previously referenced plots from the appendix (cited in Section 5.3) into the main body, ensuring the draft is more self-contained without exceeding the page limit.
>
>
> **Weakness 7:** The paper mentions that PIFA is fully differentiable, suggesting potential integration into the training stage. Have the authors explored this direction?
>
> **Reply:** Thank you for raising this interesting point. We are happy to share our preliminary findings. While PIFA ensures **lossless conversion in the forward pass**, it does **not guarantee identical gradient descent dynamics** compared to traditional low-rank training methods, as different factorizations can lead to different gradients.
>
> **Weakness 8:** How can this method integrated and benefit some recent quantization techniques like ZeroQuant-v2.
>
> **Reply:** Thank you for the suggestion. We are currently exploring this and we will provide further updates as soon as possible.

---

### Decision · Program_Chairs · 2025-05-01

**Decision:**

Accept (poster)

**Comment:**

This paper proposes an approach for low-rank pruning of LLMs, where a dense matrix W is "pruned" into two rectangular matrices AB. Noting that SVD-based factorization introduces redundancies, it introduces a more parameter-efficient factorization scheme. This in an of itself is straightforward, but the paper has additional contributions, including a GPU-friendly factorization, as well as an online data-aware reconstruction approach. The method is validated on 7B-70B LLMs where it is found to outperform baselines such as SVD-LLM.

On the plus side, the approach is sensible and clearly improves upon baselines. On the negative side, there are still significant degradations even at 80% density. And while decent speed-ups are shown for 55% density (Figure 4), but at this point the perplexity degradations make the model is pretty much unusable (e.g., 6.14PPL to 28.9PPL for LLaMA3-8B). Nevertheless, while the method is unlikely to be usable in its current form, I do think the ideas and the algorithm itself, paired with improvements over baselines, push this submission above the acceptance bar.